ecology/computational biology/behaviour

*Carcharhinus perezi*, network analysis, ontogeny, activity space, telemetry, fishing mortality

**Author for correspondence:**
Rachel T. Graham
e-mail: rachel@maralliance.org

# Movements and residency of Caribbean reef sharks at a remote atoll in Belize, Central America

Ivy E. Baremore[1], Rachel T. Graham[1], George H. Burgess[2] and Daniel W. Castellanos[1]

[1]MarAlliance, Belize City, Belize
[2]Florida Museum of Natural History, University of Florida, Dickinson Hall, Museum Road, Gainesville, FL 32611, USA

IEB, 0000-0002-7088-2219; RTG, 0000-0001-8896-1778

We investigated spatial use patterns of 77 Caribbean reef sharks (*Carcharhinus perezi*) at Lighthouse Reef Atoll, Belize over 7 years using residency patterns, kernel density (KD) estimation and network analysis. We found a high degree individual variation in spatial use of the atoll, but there were significant differences in residency and activity space between sexes, with females being overall more resident. Ontogenetic shifts in movement and residency were largely limited to females, as the residency index increased and activity space estimates decreased as females matured, while for males there was no relationship between space use or residency and size. KD analysis revealed many mature females were highly resident to discrete locations, and average activity space of the intermediate-sized sharks was significantly larger than that of the adults, but not the smallest sharks. Markov chain analyses indicated that the southwestern portion of the atoll was the most important movement corridor for all sharks. Both the Blue Hole and Half Moon Caye Natural Monuments provide some protection for larger Caribbean reef sharks; however, a gear ban on longlines on the southwestern forereef between Long Caye and the channel entrance to the Blue Hole would maximize the benefits for all sharks.

## 1. Introduction

Reef-associated sharks have faced sharp declines in recent decades, especially in the Caribbean, with abundance negatively correlated to anthropogenic factors [1–4]. Recent analysis has found that abundance and species diversity of reef sharks was significantly lower in Belize, where shark fishing occurs, than in

The Bahamas, where sharks are fully protected [5]. Despite the economic importance of reef shark species to several tourism-based economies [6–8] and fisheries [9–11], many species remain understudied in tropical seas, especially in low- and middle-income countries. Marine ecotourism is a cornerstone of the economy in Belize, and tour operators in the past have used chum to guarantee shark sightings at specific sites with several baited dives set up in Belize to attract sharks to snorkelers and divers. Feeding of spear-fished lionfish directly to sharks was also promoted in recent years as a way to 'train' sharks to eat the invasive species. Provisioning of sharks for ecotourism, as well as removals by fisheries probably affect movement and resource partitioning; for highly site-faithful reef-associated sharks, disruptions in movement patterns may have implications for the persistence of local populations [12–14].

Acoustic telemetry is a powerful tool for studying the behaviour and movements of individual sharks. Previously used for short-term (1 year or less) studies, advances in battery technology and computing power have allowed for longer-life tags to be developed and deployed in many areas around the world [15–20]. Telemetry studies are especially useful for species that are site-faithful, such as reef-associated sharks; however, few studies have investigated long-term movements of sharks with robust numbers across all life-history stages [20–24]. As most shark species undergo ontogenetic shifts in diet, habitat use and movement [25–27], it is important to take life-history stage-specific data into consideration when investigating these patterns.

The use of network analysis to describe associations among individuals and groups of elasmobranchs has recently become a popular tool among researchers [18,19,22,28–31], and many studies have found that shark species have complex behaviours that may not be discernible using observational movement analyses. Spatial networks can be used to determine the movements of animals within a region, often by direct (sighting or capture) or indirect (acoustic telemetry) observations of the individuals at discrete locations [18,19,32,33]. Coupled with a Markov chain approach, which accounts for temporal elements of movement patterns including residency periods, these approaches can identify important habitats or movement corridors, as well as sex- and/or life-history stage differences in habitat use [14,18,22]. These tools can be used to define critical habitats and movement corridors, and may be useful for analysis of the efficacy of conservation and management initiatives, such as Marine Protected Areas (MPA) [14].

Caribbean reef sharks (*Carcharhinus perezi*) are large-bodied sharks that are common throughout the tropical and subtropical waters of the western north Atlantic Ocean, and are known to be highly site-fidelic [19,34–39]. Like many reef-associated shark species, Caribbean reef sharks may spend the majority of their life history at a one reef site [34], and due to high population reductions throughout their range, were recently reassessed as 'Endangered' by the IUCN Red List of Species [40].

Network analysis of 18 Caribbean reef sharks in the US Virgin Islands showed high variation in individual space use [19], with older individuals having a preference for deeper waters. This study found strong spatial segregation of sharks by size, which the authors hypothesized was driven by territoriality and/or resource partitioning [19]; however, other authors have argued that spatial separation by reef sharks may be due to habitat-specific competition, rather than territoriality [41]. Network analysis of 20 Caribbean reef sharks in The Bahamas showed low habitat connectivity and high site fidelity [42], while network analysis of a single Caribbean reef shark in The Bahamas revealed that the individual did not use its habitat in a random manner, instead favouring several well-connected areas [28]. Studies using satellite tags showed that adult Caribbean reef sharks exhibited diel vertical migrations, while undertaking occasional deeper dives greater than 100 m [35,39]. The satellite tag studies found opposing diel vertical movement activities, with sharks in Belize using the shallow waters more at night, and those in The Bahamas using shallower waters during daylight hours [35,39].

Owing to its remote location, Lighthouse Reef Atoll (LRA) is the least accessible by fishers of the three offshore atolls in Belize (figure 1), and therefore its population of Caribbean reef sharks is probably the largest in the country. Glover's Reef Atoll is a multi-zoned marine reserve, and the use of gillnets and longlines are prohibited to the 180 m depth contour, though the forereef is a 'general use zone' and sharks are occasionally landed by hook and line. Turneffe Atoll was declared a multi-zone MPA in 2012, with enforcement beginning in 2014; however, Turneffe has a history of high fishing effort, as it is relatively easy to reach by small vessel from Belize City and the Belize Barrier Reef (figure 1). Annual scientific longline surveys have been conducted at all three atolls with varying duration since 2001 [34,43–45]. Catch per unit effort (CPUE), calculated for forereef habitat locations at LRA from 2007 to 2014 was 1.6 times higher [43] than that estimated at Glover's Reef from 2001 to 2013 [45], and nearly three times higher than that at Turneffe Atoll from 2014 to 2016 [44]. Abundance estimates for Caribbean reef sharks at LRA remained somewhat stable from 2007 to 2016, though declines from

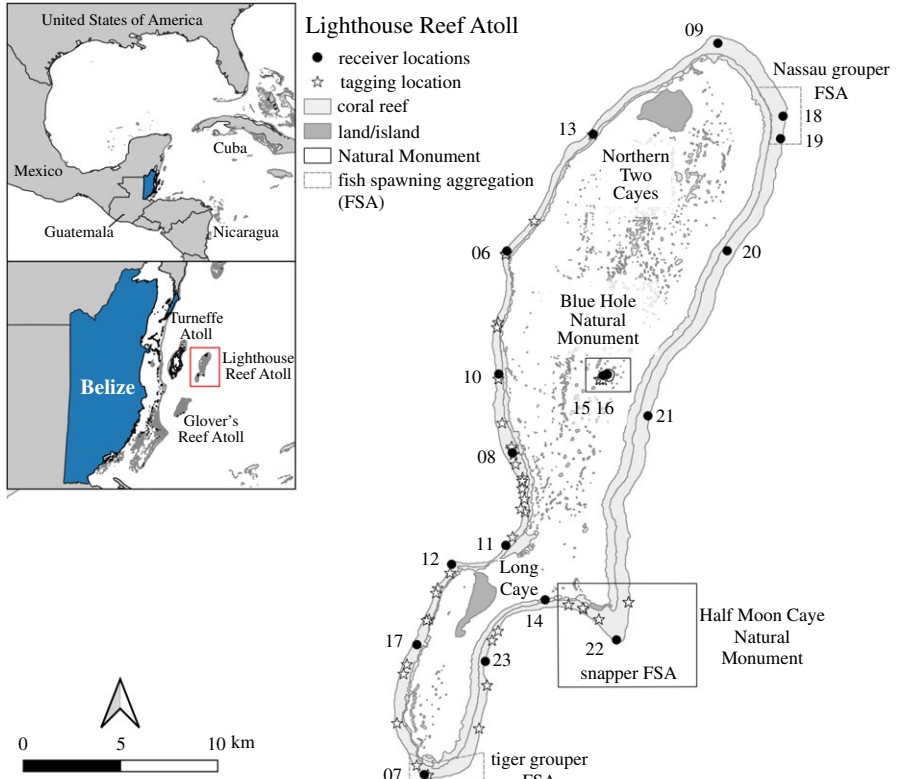

**Figure 1.** Acoustic array of receivers at LRA, Belize. Receivers, indicated by black dots and their associated numbers, were deployed continuously from 23 April 2007 to 9 May 2014, and stars indicate locations where individual Caribbean reef sharks were tagged with acoustic transmitters. Natural Monuments are no-take zones.

targeted and non-targeted small-scale fishing have been noted in recent years due to continued fishing effort to supply demand for meat in Guatemala [43,46,47].

The objectives of this study are to examine the spatial use of different life stages of Caribbean reef sharks at LRA, and to examine how ontogenetic patterns in space use and movements help to drive the partitioning of the atoll. These results will help to determine the degree of vulnerability of the sharks to overexploitation to better guide protected areas managers in conservation measures within the atoll's protected areas and the Government of Belize in their management of shark fisheries in both distribution and effort.

# 2. Material and methods

## 2.1. Site description

LRA is the most remote of the three atolls that are part of the Mesoamerican Barrier Reef in Belize, Central America (figure 1). The atoll is approximately 300 km$^2$ in area and is comprised of a fringed forereef surrounding a shallow lagoon, encompassing mostly sand and patch reef habitat. The atoll (figure 1) hosts several fish spawning aggregation (FSA) sites for snappers (*Lutjanus cyanopterus* and *L. jocu*), groupers (*Epinephelus striatus* and *Mycteroperca tigris*), and several species of jacks (Carangidae). Grouper spawning season generally occurs in the region from November to February, and there is a closed season for Nassau grouper in Belize from 1 December to 31 March. A popular dive location with important fishing grounds for finfish, conch and lobster, LRA is famous for the Great Blue Hole and Half Moon Caye wall dive sites, both of which are World Heritage Sites and Natural Monuments declared in 1996, benefiting from an enforced no-take protected status. Both sites are known globally to divers for predictable shark encounters with Caribbean reef sharks.

## 2.2. Acoustic array

A total of 18 VEMCO VR2-W acoustic receivers (Amirix Systems, Nova Scotia, Canada) were stationed at LRA, Belize from 23 April 2007 to 9 May 2014 (figure 1). Receivers were non-overlapping and

positioned between 8 and 25 m depth, in forereef and reef pass habitats in several locations around the entire atoll, and average distance between receivers was 5.5 km. Receivers were removed briefly (less than 24 h) every three to six months for battery changes and data retrieval. A range detection study conducted as part of the project yielded ranges of up to 250 m for reception of tags (RT Graham 2007, unpublished data).

Caribbean reef sharks were captured by standardized longline on various forereef locations around the atoll and in the Blue Hole [48] during three seasonal sampling periods from 2007 to 2009 (figure 1). Longline sets were comprised of a 500 m floating rope main line with 50, 4 m gangions attached with a clip at regular intervals between 11 equally spaced floats. Gangions were stainless steel braided 1/16 leader terminating in a 16/0 offset triple strength stainless steel circle hook. Soak time was limited to 90 min to minimize stress-induced mortality, and hooks were removed when possible prior to release. Upon capture, all sharks were measured in-water for precaudal (PCL) and total lengths (TL, cm), tagged externally with a conventional tag, and sexed. Sharks considered to be in excellent condition were placed in tonic immobility alongside the vessel, and were surgically implanted with an acoustic transmitter (Vemco V16-6 L coded acoustic transmitter, 69 kHz, 90 s blanking interval, estimated tag life = 1877 d), via a 4–5 cm incision into the peritoneal cavity made using a sterile #10 surgical scalpel blade. Following insertion of the tag, incisions were closed using Ethicon 5–0 cutting edge needle and braided silk sutures. Tagging procedures were typically under 10 min, and sharks were revived before they were released.

## 2.3. Data analyses

Sharks detected at the array for fewer than 5 days total were excluded from analyses due to lack of data, and single detections per day at a receiver were eliminated to remove potential false detections. Sharks were categorized into three life-history stages based on sex and size: size class A were juvenile sharks (less than 150 cm TL for females and less than 130 cm for males); size class B were subadults (150–180 cm TL for females and 130–160 cm TL for males); and size class C were mature (greater than 180 cm TL for females, and greater than 160 cm TL for males) (35).

Residency was determined by daily occurrence (a shark detected at the array at least twice per day). A daily residency index was calculated as the total days each shark was detected at the array divided by the number of days between the date it was tagged and the last date expected due to the lifespan of the acoustic tag (RI = number of days detected/1877). As the RI based on tag life is a very conservative estimate and probably variable, a second residency index ($RI_{max}$) was calculated based on the last day the shark was detected at the array [49]. In cases where the tag outperformed its estimated tag life, the number of expected days was adjusted according to the last day detected or the end of the study period, whichever was first, and sharks that were reported as captured by fishers were given the capture date as the last expected day. A roaming index, which is the number of receivers visited by each shark divided by the number of receivers at the array, was calculated to assess the extent of movement by individual sharks within the array [50]. Generalized linear models (GLM) were used to investigate how size and sex influenced the RI and roaming index of Caribbean reef sharks at LRA. If overdispersion was detected, a GLM with a negative binomial error distribution was used, otherwise a GLM with a Gaussian error distribution was applied [51], and the full models included sex, TL and the interaction of the two factors.

To assess trends in residency over the study period, RI was also calculated by month for each individual using the total number of days detected each month divided by the number of days in the month. Monthly RI values were averaged and plotted by sex and size class by month, and also for the entire monitoring period by sex. Because the tag life of more than half of tagged sharks terminated before the end of the monitoring period, average monthly RI was calculated only through the end of 2012. Differences in monthly RI between sexes were investigated using a Mann–Whitney U-test, and among size classes with a Kruskal–Wallis rank sum test, with a post hoc multiple comparison Kruskal–Wallis performed if differences were significant [52]. Mixed effects models (GLMM) were used to investigate factors influencing monthly RI of Caribbean reef sharks over the monitoring period. Individual (ID) and year were included in the model as random factors to account for repeated measures [51]. Models were analysed for multi-collinearity by calculating variance inflation factors (VIF). Final model selection was based on Akaike information criteria (AIC) [51,53], and the dredge function (package MuMIn) [54] was used to generate a suite of models for comparison. Candidate models were tested against the null model using maximum likelihood. The global model was: Monthly residency~Month (integer) + Sex + TL + [1 | ID] + [1 | Year].

### 2.3.1. Core activity space analyses

Kernel density (KD) estimates for individual sharks were analysed using a bivariate normal kernel function [55]. The KD calculation uses a kernel method to estimate the KD using relocation (detection) data [56], and a smoothing parameter was chosen based on visual choice after several trials [55]. Because the receiver array was relatively large (approx. 300 km$^2$) and non-overlapping, direct movement between receivers could not be determined; therefore, a 30 min time step for each individual was used. Total (95% KD) and core activity space (50% KD) was examined for each shark and by sex and size class. The total activity space (95% KD) was larger than the available habitat of LRA for some individuals; therefore, only core activity space (50% KD) was reported. Core activity spaces were estimated only for sharks that had sufficient relocations at more than three receivers. A generalized linear model with a Poisson error distribution was used to determine whether there was a relationship between shark TL and core activity space size, with the 50% KD log transformed to normalize the data. Model selection was based on AIC selection criteria, with the final model having the lowest AIC [51]. Generalized additive models (GAM) were considered, but not reported in final analyses after examination of residuals and AIC values.

### 2.3.2. Markov chain analyses

To determine sex and life-history stage differences in movements within the atoll, spatial empirically derived Markov chain (EDMC) analyses of detection data were performed [18]. Based on network analysis, EDMC models the temporal dimensions of movements, including residency and absence periods [18]. The Markov chain approach requires several assumptions about the movements of sharks and study area characteristics [18], all of which were met by the dataset. Receivers that were in close proximity (less than 1 km apart) were combined to avoid confounding detections: data for the two receivers at the Blue Hole (15 and 16) were combined, as were receivers 18 and 19 at the northeastern point (figure 1) to create a network of 16 non-overlapping 'nodes' that were between 3 and 10 km apart. Assuming an average straight-line swimming speed between 4 and 12 km h$^{-1}$ [57], movement data were aggregated temporally into 1 h intervals to reduce computation time [41] and to allow for transitions between receivers. Each hourly interval with an associated detection at a receiver was assigned a 1 (present) and the hourly intervals without a detection at a receiver was assigned a 0 (absent). Matrices of movement counts were computed for the hourly time step, so that the square matrix contained movements from each receiver to itself (residency periods) as well as to the absent state (transition period). Eigenvector centralities of the transition matrices were calculated as a proxy for the probability of a shark being at a given receiver [18,22]. Estimation of the dominant eigenvector was calculated using the power method and EDMC analyses were carried out in R using code provided by Stehfest *et al.* [18].

All statistical analyses were performed in R [58]: the igraph [59] and ggnet [60] packages were used for network analyses and visualizations in R. AdehabitatHR [55] was used to estimate KD; sp, mapproj, and maptools packages were used to make shapefiles [61–63]; and the tidyverse packages were used for data tidying, analysis and plotting [64]. Maps were made using QGIS 3.16.0 [65] and R.

## 3. Results

A total of 87 (46 females and 41 males) Caribbean reef sharks were tagged over a period of three sampling events from April 2007 to May 2009 (electronic supplementary material, table S1). Of these sharks, four were never detected at the array and another six were detected for fewer than five total days, leaving 77 (39 females and 38 males) to be included in further analyses (table 1): 24 (14 females, 10 males) were in size class A, 14 (5 females, 9 males) were in size class B, and 39 (20 females, 19 males) were in size class C (table 1).

The number of days between first and last detections ranged from 7 to 1973 days for the 77 Caribbean reef sharks tracked at the array (table 1, electronic supplementary material, figure S1). Residency during the study period, as measured by the expected life of the tags, was relatively low overall (mean RI = 0.22 ± 0.21 s.d.), with 89% of sharks having RI < 0.50 (table 1). Residency as measured by date of last detection was higher (mean RI$_{max}$ = 0.44 ± 0.29 s.d.), and 39% ($n$ = 30) had RI$_{max}$ values > 0.50 (table 1 and figure 2a). More than half (56%) of tagged sharks had a roaming index ≤ 0.50, indicating that the sharks had variable movement patterns throughout the atoll (figure 2a). Residency increased for females with size, and males maintained similar RI as they grew (figure 2b). Results from GLM analyses indicated that RI was influenced more by sex than size of the shark, though both TL and sex

**Table 1.** Summary of detections and size data for each of the 77 tagged Caribbean reef sharks (*Carcharhinus perezi*) detected more than five total days at the acoustic array at LRA, Belize, from 2007 to 2014. The number of receivers visited, residency indices (RI), roaming index, and total days detected at the array are reported for all sharks, and core activity space estimates (50% KD) are reported for the 68 Caribbean reef sharks for which there were sufficient relocations among receivers to calculate KD. The table is sorted by sex and the number of days detected.

| ID | sex | size class | date tagged | TL (cm) | days detected | receivers visited | RI | $RI_{max}$ | roaming index | 50% KD (km$^2$) |
|----|-----|-----------|-------------|---------|---------------|-------------------|-----|-----------|---------------|------------------|
| 79 | F | A | 30 Apr 2009 | 113.5 | 29 | 4 | 0.015 | 0.071 | 0.222 | 131.793 |
| 61 | F | C | 1 May 2009 | 212.5 | 39 | 1 | 0.021 | 0.907 | 0.056 | |
| 53 | F | B | 23 Nov 2007 | 154.5 | 40 | 8 | 0.021 | 0.159 | 0.444 | 254.556 |
| 5 | F | B | 24 Apr 2007 | 152 | 50 | 8 | 0.089 | 0.089 | 0.444 | 222.006 |
| 84 | F | A | 3 May 2009 | 145 | 103 | 10 | 0.055 | 0.098 | 0.556 | 24.745 |
| 29 | F | C | 28 Nov 2007 | 208 | 116 | 3 | 0.062 | 0.126 | 0.167 | |
| 69 | F | C | 9 Jun 2008 | 188 | 117 | 2 | 0.062 | 0.975 | 0.111 | |
| 73 | F | A | 10 Jun 2008 | 124 | 142 | 7 | 0.076 | 0.594 | 0.389 | 0.316 |
| 80 | F | A | 30 Apr 2009 | 144.5 | 169 | 14 | 0.090 | 0.411 | 0.778 | 1.587 |
| 33 | F | C | 27 Nov 2007 | 195 | 171 | 12 | 0.091 | 0.187 | 0.667 | 81.087 |
| 25 | F | A | 24 Apr 2007 | 125.5 | 229 | 9 | 0.122 | 0.250 | 0.500 | 20.334 |
| 58 | F | A | 5 May 2009 | 144.5 | 237 | 11 | 0.126 | 0.587 | 0.611 | 0.124 |
| 20 | F | C | 23 Apr 2007 | 220 | 273 | 4 | 0.145 | 0.344 | 0.222 | 0.797 |
| 6 | F | B | 4 May 2007 | 150.5 | 278 | 2 | 0.148 | 0.955 | 0.111 | 0.003 |
| 18 | F | C | 1 May 2007 | 212 | 305 | 6 | 0.162 | 0.645 | 0.333 | 2.162 |
| 7 | F | C | 27 Apr 2007 | 226 | 307 | 3 | 0.163 | 0.865 | 0.167 | |
| 55 | F | C | 6 May 2009 | 206 | 325 | 7 | 0.173 | 0.405 | 0.389 | 0.536 |
| 70 | F | A | 10 Jun 2008 | 117 | 341 | 10 | 0.181 | 0.553 | 0.556 | 0.361 |
| 50 | F | A | 23 Nov 2007 | 131.5 | 350 | 12 | 0.186 | 0.207 | 0.667 | 3.331 |
| 27 | F | A | 10 Jun 2008 | 99 | 400 | 6 | 0.213 | 0.932 | 0.333 | 0.049 |
| 82 | F | B | 4 May 2009 | 153 | 431 | 13 | 0.229 | 0.372 | 0.722 | 6.642 |

(*Continued.*)

**Table 1.** (*Continued.*)

| ID | sex | size class | date tagged | TL (cm) | days detected | receivers visited | RI | RI$_{max}$ | roaming index | 50% KD (km$^2$) |
|---|---|---|---|---|---|---|---|---|---|---|
| 14 | F | A | 3 May 2007 | 113 | 471 | 8 | 0.251 | 0.329 | 0.444 | 3.319 |
| 59 | F | C | 5 May 2009 | 193.5 | 485 | 4 | 0.258 | 0.362 | 0.222 | 0.458 |
| 47 | F | A | 24 Nov 2007 | 113.5 | 490 | 8 | 0.261 | 0.480 | 0.444 | 1.152 |
| 72 | F | C | 1 May 2009 | 201 | 492 | 4 | 0.262 | 0.713 | 0.222 | 1.368 |
| 49 | F | A | 24 Nov 2007 | 113 | 534 | 17 | 0.284 | 0.438 | 0.944 | 57.819 |
| 57 | F | C | 6 May 2009 | 232.5 | 555 | 5 | 0.295 | 0.690 | 0.278 | 0.003 |
| 46 | F | C | 27 Nov 2007 | 218 | 573 | 13 | 0.305 | 0.450 | 0.722 | 37.559 |
| 42 | F | C | 25 Nov 2007 | 213 | 624 | 5 | 0.332 | 0.530 | 0.278 | 0.082 |
| 2 | F | C | 26 Apr 2007 | 201 | 693 | 5 | 0.369 | 0.852 | 0.278 | 0.002 |
| 86 | F | C | 11 May 2009 | 181 | 705 | 8 | 0.375 | 0.770 | 0.444 | 2.738 |
| 77 | F | A | 29 Apr 2009 | 108 | 919 | 17 | 0.489 | 0.869 | 0.944 | 2.741 |
| 64 | F | B | 8 May 2009 | 155.5 | 972 | 9 | 0.517 | 0.575 | 0.500 | 17.714 |
| 10 | F | A | 4 May 2007 | 133.5 | 1115 | 13 | 0.593 | 0.673 | 0.722 | 1.153 |
| 13 | F | C | 24 Apr 2007 | 206 | 1295 | 6 | 0.544 | 0.544 | 0.333 | 1.709 |
| 81 | F | C | 30 Apr 2009 | 212.5 | 1566 | 9 | 0.833 | 0.856 | 0.500 | 0.414 |
| 44 | F | C | 27 Nov 2007 | 212 | 1567 | 3 | 0.834 | 0.884 | 0.167 | |
| 8 | F | C | 27 Apr 2007 | 223 | 1732 | 2 | 0.887 | 0.887 | 0.111 | |
| 67 | F | C | 9 Jun 2008 | 187 | 1973 | 10 | 0.915 | 0.915 | 0.556 | 2.191 |
| 41 | M | C | 24 Nov 2007 | 183 | 7 | 3 | 0.004 | 0.037 | 0.167 | 12.173 |
| 54 | M | C | 23 Nov 2007 | 186 | 10 | 1 | 0.005 | 0.006 | 0.056 | |
| 24 | M | B | 24 Apr 2007 | 147 | 16 | 4 | 0.009 | 0.063 | 0.222 | 50.980 |
| 9 | M | B | 25 Apr 2007 | 134 | 21 | 4 | 0.011 | 0.389 | 0.222 | 69.579 |
| 34 | M | C | 28 Nov 2007 | 178.5 | 40 | 4 | 0.021 | 0.132 | 0.222 | 96.753 |

(*Continued.*)

**Table 1.** (*Continued.*)

| ID | sex | size class | date tagged | TL (cm) | days detected | receivers visited | RI | RI$_{max}$ | roaming index | 50% KD (km$^2$) |
|----|-----|-----------|-------------|---------|---------------|-------------------|-----|-----------|---------------|------------------|
| 75 | M | A | 29 Apr 2009 | 103 | 45 | 12 | 0.024 | 0.039 | 0.667 | 41.066 |
| 22 | M | A | 22 Apr 2007 | 119 | 47 | 1 | 0.025 | 0.065 | 0.056 | |
| 16 | M | C | 4 May 2007 | 198.5 | 54 | 3 | 0.029 | 0.684 | 0.167 | 1.011 |
| 37 | M | B | 27 Nov 2007 | 158.5 | 58 | 6 | 0.031 | 0.044 | 0.333 | 53.623 |
| 30 | M | C | 28 Nov 2007 | 196 | 69 | 4 | 0.037 | 0.116 | 0.222 | 11.778 |
| 62 | M | B | 7 May 2009 | 137.5 | 81 | 3 | 0.043 | 0.216 | 0.167 | |
| 87 | M | C | 11 May 2009 | 196.5 | 90 | 5 | 0.048 | 0.201 | 0.278 | 4.133 |
| 28 | M | C | 28 Nov 2007 | 187 | 103 | 5 | 0.055 | 0.367 | 0.278 | 15.344 |
| 17 | M | A | 3 May 2007 | 110 | 127 | 4 | 0.068 | 0.143 | 0.222 | 0.471 |
| 36 | M | C | 27 Nov 2007 | 193 | 131 | 5 | 0.068 | 0.068 | 0.278 | 34.802 |
| 52 | M | B | 22 Nov 2007 | 136 | 181 | 13 | 0.096 | 0.192 | 0.722 | 65.855 |
| 3 | M | C | 4 May 2007 | 183 | 192 | 13 | 0.102 | 0.169 | 0.722 | 26.677 |
| 4 | M | C | 27 Apr 2007 | 196.5 | 232 | 8 | 0.123 | 0.536 | 0.444 | 0.093 |
| 23 | M | A | 22 Apr 2007 | 117 | 233 | 10 | 0.124 | 0.270 | 0.556 | 78.348 |
| 1 | M | C | 24 Apr 2007 | 193.5 | 239 | 11 | 0.127 | 0.177 | 0.611 | 7.862 |
| 51 | M | B | 23 Nov 2007 | 159 | 249 | 16 | 0.132 | 0.144 | 0.889 | 44.398 |
| 15 | M | A | 3 May 2007 | 108 | 251 | 5 | 0.134 | 0.374 | 0.278 | 3.475 |
| 56 | M | C | 5 May 2009 | 175 | 292 | 13 | 0.155 | 0.187 | 0.722 | 43.719 |
| 68 | M | A | 11 May 2009 | 108.5 | 298 | 11 | 0.159 | 0.937 | 0.611 | 30.302 |
| 85 | M | C | 11 May 2009 | 167 | 327 | 15 | 0.174 | 0.191 | 0.833 | 9.204 |
| 11 | M | B | 1 May 2007 | 156 | 391 | 11 | 0.208 | 0.250 | 0.611 | 3.905 |
| 12 | M | B | 4 May 2007 | 133.5 | 493 | 13 | 0.262 | 0.410 | 0.722 | 20.443 |
| 83 | M | C | 6 May 2009 | 172.5 | 496 | 10 | 0.264 | 0.428 | 0.556 | 5.721 |

(*Continued.*)

**Table 1.** (*Continued.*)

| ID | sex | size class | date tagged | TL (cm) | days detected | receivers visited | RI | RI$_{max}$ | roaming index | 50% KD (km$^2$) |
|---|---|---|---|---|---|---|---|---|---|---|
| 48 | M | A | 24 Nov 2007 | 116 | 507 | 9 | 0.270 | 0.450 | 0.500 | 3.071 |
| 74 | M | A | 29 Apr 2009 | 104.5 | 536 | 14 | 0.285 | 0.479 | 0.778 | 0.555 |
| 65 | M | A | 11 May 2009 | 110.5 | 558 | 8 | 0.297 | 0.669 | 0.444 | 0.590 |
| 38 | M | C | 27 Nov 2007 | 183.5 | 615 | 5 | 0.327 | 0.357 | 0.278 | 2.886 |
| 71 | M | C | 2 May 2009 | 178.5 | 623 | 9 | 0.331 | 0.341 | 0.500 | 6.098 |
| 66 | M | B | 10 Jun 2008 | 146 | 633 | 8 | 0.337 | 0.337 | 0.444 | 0.364 |
| 78 | M | C | 30 Apr 2009 | 202.5 | 634 | 14 | 0.337 | 0.553 | 0.778 | 16.252 |
| 63 | M | C | 2 May 2009 | 184.5 | 761 | 9 | 0.405 | 0.562 | 0.500 | 5.173 |
| 76 | M | A | 29 Apr 2009 | 110.5 | 837 | 11 | 0.445 | 0.788 | 0.611 | 3.843 |
| 60 | M | C | 7 May 2009 | 188 | 1115 | 8 | 0.593 | 0.806 | 0.444 | 1.593 |

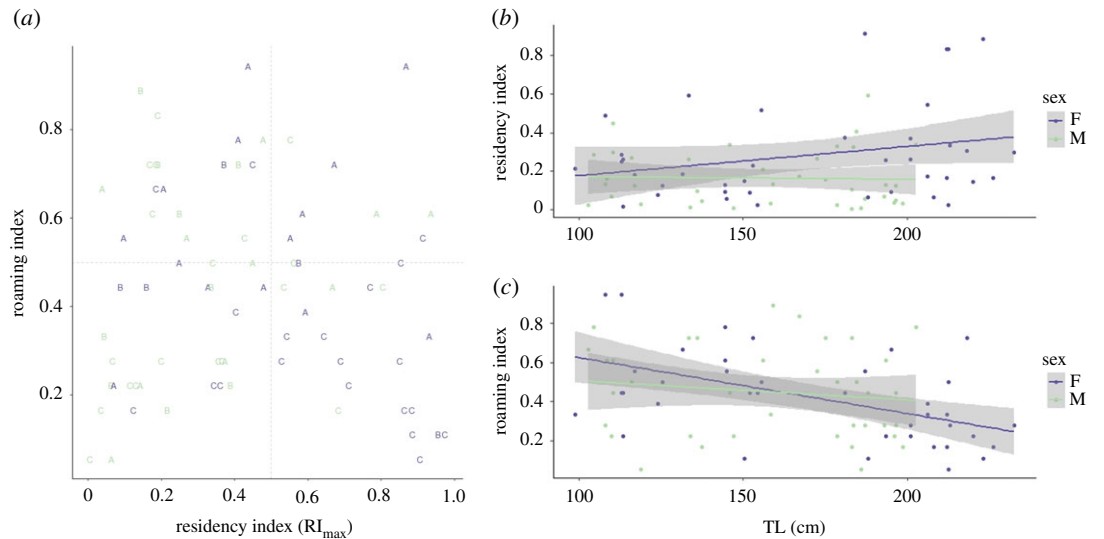

**Figure 2.** Relationships between (*a*) maximum residency (RI$_{max}$) and roaming index by sex and size class (A, juvenile; B, subadult; C, adult); (*b*) residency (RI) and shark total length (TL); and (*c*) roaming index and shark TL showing decreased roaming and increased residency by female Caribbean reef sharks with increasing size at LRA, Belize.

**Table 2.** Results of GLM analyses showing factors influencing residency index (RI) and roaming index of Caribbean reef sharks at LRA, Belize. Significant effects ($p < 0.05$) are in italics.

|  | estimate | s.e | t-value | p-value |
|---|---|---|---|---|
| *RI* | | | | |
| intercept | 0.1331 | 0.1094 | 1.2170 | 0.2275 |
| TL | 0.0009 | 0.0006 | 1.4330 | 0.1562 |
| sex | −0.1088 | 0.0468 | −2.3250 | *0.0228* |
| *roaming index* | | | | |
| intercept | 0.7845 | 0.1093 | 7.1760 | *0.0000* |
| TL | −0.0021 | 0.0007 | −3.2620 | *0.0017* |

were selected for the final model (table 2). The roaming index significantly decreased for females with increasing TL, and showed no trend for males (figure 2*c*), with results from GLMs indicating that the roaming index was significantly influenced by size (table 2). Most sharks (61%) with roaming indices less than 0.5 were mature, and of the 19 sharks with RI$_{max}$ greater than 0.50 and roaming index less than 0.50 (i.e. site- and array-faithful), 63% were mature females (figure 2*a*).

Monthly RI was overall higher for females (mean = 0.63 ± 0.35 s.d.) than males (mean = 0.43 ± 0.33 s.d.) (Mann–Whitney *U*, $p < 0.0001$), and monthly RI was significantly different among size classes (Kruskal–Wallis rank sum test, $p < 0.0001$) (electronic supplementary material, figure S2*a,b*). The smallest and largest size classes had similar mean monthly RI (size class *A* = 0.55 ± 0.37, size class *C* = 0.58 ± 0.35 s.d.; post hoc multiple comparison Kruskal–Wallis $p > 0.05$), while sharks in the intermediate size class, B, had the smallest mean monthly RI (0.41 ± 0.31) and monthly RI was significantly smaller than that of size classes A and C (post hoc Kruskal–Wallis $p < 0.05$). Results of mixed model analysis indicated that month and sex had the biggest influence on monthly RI, followed by the interaction term of sex and month, then TL (table 3).

## 3.1. Atoll use

### 3.1.1. Core activity space

KD analysis was conducted for the 68 sharks that were detected at more than three receivers and with sufficient relocations to calculate 50% KD. Core activity space size was variable among individuals

**Table 3.** Results of GLMM model selection showing shark sex and month were the most influential factors for monthly residency index (RI) of Caribbean reef sharks at LRA, Belize. The best model as determined by the model selection criteria is in italics.

| factors | d.f. | logLik | AICc | ΔAIC | w |
|---|---|---|---|---|---|
| *Sex + Month* | *6* | *−268.138* | *548.3* | *0* | *0.422* |
| Sex + Month + Sex*Month | 7 | −267.608 | 549.3 | 0.95 | 0.262 |
| Sex + Month + TL | 7 | −267.93 | 549.9 | 1.6 | 0.19 |
| Sex + Month + TL + Sex*Month | 8 | −267.397 | 550.9 | 2.55 | 0.118 |
| Month | 5 | −273.55 | 557.1 | 8.81 | 0.005 |
| Month + TL | 6 | −272.867 | 557.8 | 9.46 | 0.004 |
| Sex | 5 | −278.574 | 567.2 | 18.86 | 0 |
| Sex + TL | 6 | −278.366 | 568.8 | 20.46 | 0 |
| – | 4 | −283.938 | 575.9 | 27.58 | 0 |
| TL | 5 | −283.258 | 576.5 | 28.23 | 0 |

(avg. 24.3, range 0.001–254.6 km$^2$) (figure 3 and table 1). Visual analysis showed some partitioning of the atoll by size class and sex: large, mature females (size class C) were more commonly found in the southeastern half of the atoll, and the smaller females (size classes A and B) mostly used the middle and lower portion of LRA (figure 3a–c). Males' activity spaces were patchier than the females', and mature males were the main group that used the northeastern side of the atoll (figure 3d–f). The subadult size class had the largest average activity space (mean 62.3 km$^2$ ± 82.2 s.d.), which was significantly larger than the activity space size of the adult size class (mean 13.5 km$^2$ ± 23.0 s.d., post hoc Kruskal–Wallis $p < 0.05$), but not the juveniles' (mean 17.8 km$^2$ ± 32.5 s.d., $p > 0.05$). Of the 20 mature females tracked over the course of the study, KD could not be calculated for 6 (30%) due to lack of movements among more than three receivers.

Sex was the best predictor for activity space size, with log-transformed 50% KD estimates decreasing for females with increasing size (TL), while activity space showed no linear trend for males with size (table 4; electronic supplementary material, figure S3). The final selected model included only sex (table 4). Lack of movement by most sharks among more than a few receivers negated analysis of activity space by month or by months aggregated into seasons.

### 3.1.2. Markov chain analyses

Eigenvalue centrality ranks indicated that sharks in all size classes were most likely to be spatially absent from the array, with probabilities ranging from 0.72 to 0.95, probably signifying that animals spent most of their time outside receivers' ranges (electronic supplementary material, table S2). Outside the absent states, sharks showed some spatial differences among sexes and size classes, though all sharks tended to use the southwestern portion of the atoll most frequently (figure 4). Both mature females and subadult males were found to prefer the Blue Hole site (electronic supplementary material, table S2), although there were no transitions into or away from the Blue Hole estimated for the hourly time step for either group (figure 4c–e). Juveniles and subadults probably used the lagoon areas for transitions among receivers more than the adult size classes (figure 4). Mature females' highest transition probabilities were between the receivers on either side of a channel between Half Moon and Long Cayes, while mature males mostly moved among the receivers on the western side of the atoll and to the Blue Hole.

## 4. Discussion

This study revealed individual variation in movement and spatial partitioning among Caribbean reef sharks at LRA, Belize. Analyses of residency, space use and movement patterns indicated that Caribbean reef sharks had ontogenetic shifts in atoll use, with individuals from the intermediate size class (B) having significantly lower monthly RI and larger core use areas than adults (size class C). Analyses of trends in overall residency, roaming index and core use area indicated that sharks became

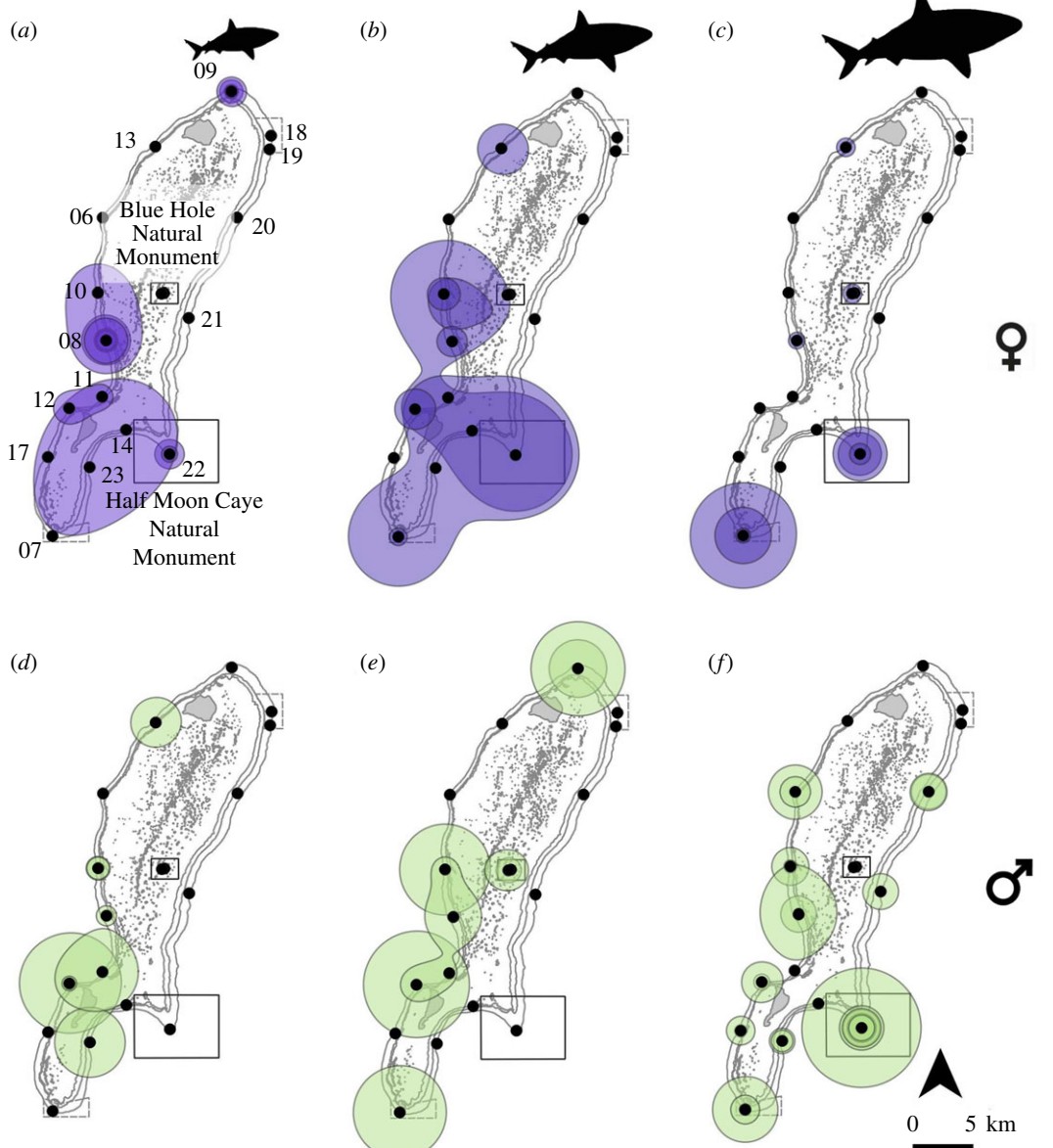

**Figure 3.** KD home activity spaces for Caribbean reef sharks plotted by sex and life-history stage at LRA, Belize: (*a*) females, size class A; (*b*) females, size class B; (*c*) females, size class C; (*d*) males, size class A; (*e*) males, size class B; (*f*) males, size class C. Females' activity spaces are in purple and males in green, black dots represent receiver locations and the numbers represent station identifications.

**Table 4.** Results of GLM analyses showing the activity space (log-transformed 50% KD). Significant effects ($p < 0.05$) are in italics.

| log(KD) | estimate | s.e | *t*-value | *p*-value |
| --- | --- | --- | --- | --- |
| intercept | 0.2156 | 0.1857 | 1.161 | 0.2498 |
| *sex* | *0.6848* | *0.2589* | *2.645* | *0.0102* |

more resident and had smaller home ranges as they matured, but that these trends were mostly restricted to females. The juvenile and subadult size classes were also more likely to use movement corridors over interior lagoon habitats. The larger activity space and low monthly residency index among subadult Caribbean reef sharks may indicate that the mid-sized sharks disperse to other locations to reduce

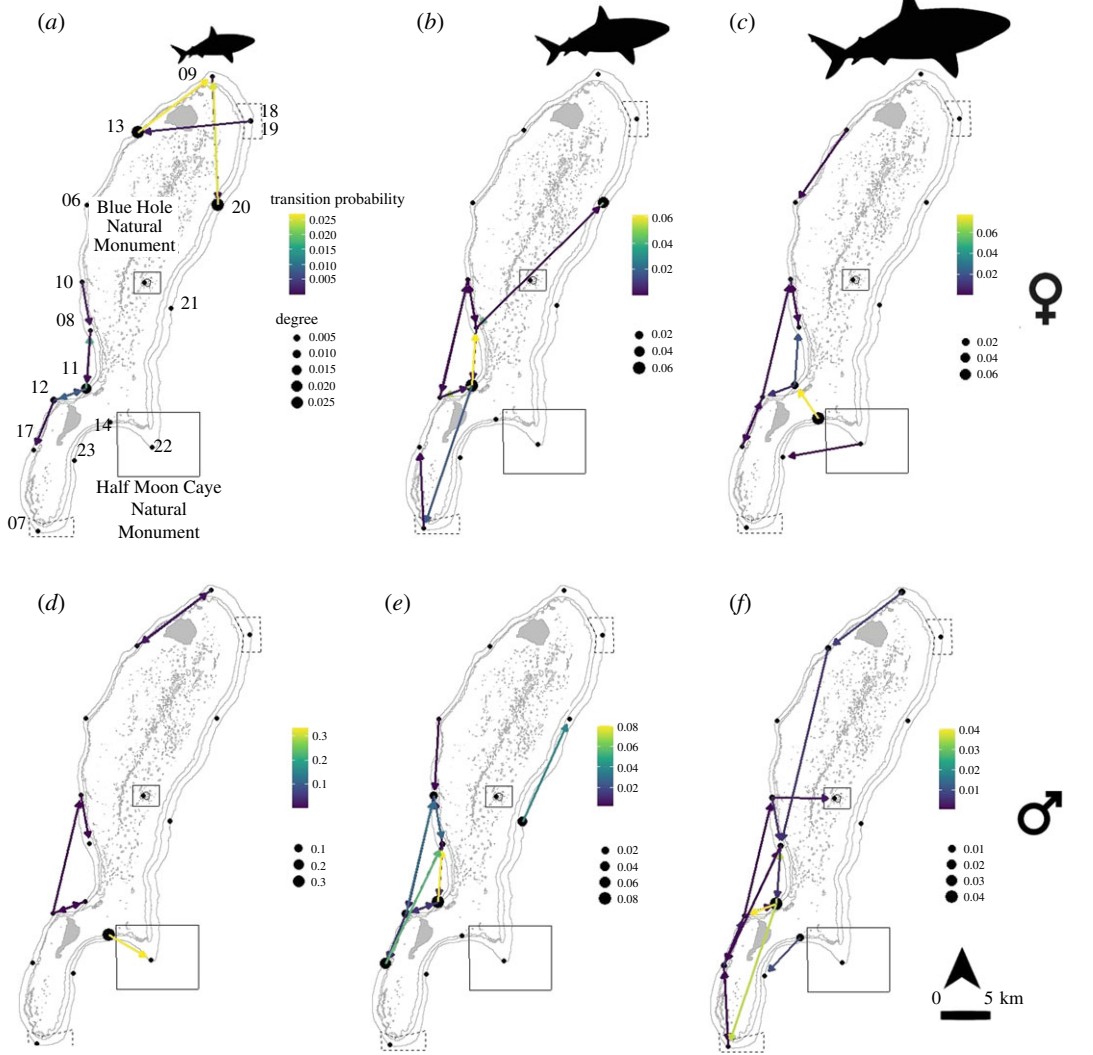

**Figure 4.** Movement transition probabilities of Caribbean reef sharks among receivers at LRA, Belize, calculated by sex and life-history stage: (*a*) females, size class A; (*b*) females, size class B; (*c*) females, size class C; (*d*) males, size class A; (*e*) males, size class B; (*f*) males, size class C. Arrow colour is indicative of probability, and receiver node size is degree, calculated from Markov chain analyses of transmitter data.

competition [66]. Six of the acoustically tagged sharks were detected at acoustic arrays more than 90 km from LRA, and two were captured by fishers in southern Quintana Roo, Mexico (RT Graham 2016, unpublished data). Of these animals, four were juveniles or subadults at the time of tagging, and two of the juveniles detected away from the array were not detected at LRA afterwards. Future genetic studies should be undertaken to determine the relatedness of the sharks at LRA, and to examine the possibility of male-mediated gene flow for Caribbean reef sharks at different reef sites [67].

Mature females showed the most specialization in movement and residency patterns, having in general the highest residency, lowest roaming index and smallest core use areas. KD analyses indicated that the area around the Half Moon Caye Natural Monument and southern reef fish FSA was among the most important for mature females in terms of core use. The Half Moon Caye reef wall is a highly productive site, and hosts snapper spawning aggregations during summer months (May–August), while the northern and southern sites are grouper spawning sites during winter months (November–March). It is reasonable to assume that the largest females are dominant in their preferred activity spaces. The energetic cost of reproduction is much higher for females than males, and therefore these females may exhibit preference for the spawning aggregation sites because they provide a predictable and high-nutrient source of food [68,69]. Mature males, while having similarly

small average home ranges to the females, showed somewhat more dispersion throughout the atoll, perhaps to maintain their overall larger range in order to gain more mating opportunities and decrease inbreeding [70], or simply to avoid competition with the dominant females during non-mating periods. Future investigations of social network structure for this population will help to elucidate these suppositions, as other studies have found significant leadership patterns by sex [71], multi-year, spatially assorted social communities in grey reef sharks (Carcharhinus amblyrhynchos) [72] and social groupings of other shark species by sex and size that may vary seasonally and over distinct habitats and locations [33,73,74].

The southwestern forereef habitat between Long Caye (receivers 11 and 12) and entrance to the channel for the Blue Hole (receiver 10) was the most important movement corridor for all sizes and sexes of Caribbean reef sharks at LRA, despite differences in core activity spaces. The western forereef habitat of LRA has a steeper drop-off than the eastern side, and generally hosts the highest abundance of sharks of the atoll [43]. Although remote, LRA can be heavily targeted by shark fishers during the months leading up to Easter (December–March) due to high demand for fish during the Lenten season, especially from Guatemala. A constantly occupied ranger station on Half Moon Caye and daily patrols at Blue Hole largely discourage shark fishing in the MPAs. Of the 77 sharks tracked during this study, 6.5% were reported landed by fishers (three at LRA), and more were captured as verified by photos of external tags but were unreported by fishers and therefore could not be identified (RT Graham 2017, personal observation). The western side of the atoll is sheltered from the predominant northeasterly winds, and therefore the majority of fishing effort is focused on this side, and fishers are known to use Northern Two Cayes and a small caye south of Long Caye as base camps. A ban on shark fishing gear (i.e. longline) along this important movement corridor would offer protection for all sexes and life-history stages of Caribbean reef sharks while allowing continuation of traditional fishing of lobster, conch and finfish.

Overall, Caribbean reef sharks are known for high site fidelity throughout their range [19,34–36,38,39,45]. Caribbean reef sharks tagged at Glover's Reef Atoll, Belize had higher residency index (43% at Glover's versus 22% in this study), though differences in the residency index was not observed between juveniles and adults [34]. These differences are probably due to the scale and period of monitoring, as only eight of the 31 sharks monitored at Glover's Reef were mature and the monitoring period was shorter than that of the present study. In Brazil, juvenile Caribbean reef sharks showed high fidelity to their original tagging site, estimated short-term activity spaces were less than 1 km$^2$, and ontogenetic changes in movements were postulated but not empirically observed [36].

Results from this study showed that females became more resident and site-fidelic as they grew and matured. Targeted fishing mortality generally removes the largest animals from a population first, and at LRA the truncation of the size frequency has been noted [43]; however, sharks may become resilient to recapture, and social networks of sharks have shown to be robust to the removal of highly connected individuals [13]. Further analyses of social networks using simulated bootstrapping methods could provide insight into the effects of overfishing, but it is likely that the loss of the large females would be detrimental to the population of Caribbean reef sharks at LRA. Future studies should focus on the effects of ongoing fishing mortality and include genetic population structure analysis to determine how relatedness of individuals may also influence the resource partitioning of this economically and socially important population of sharks.

Ethics. Research was conducted under annually renewed research permits with the Belize Fisheries Department. Permit numbers: 0003-07; 0005-08; 0001-09; 0012-10; 0006-11; 0009-12; 0011-13.

Data accessibility. Data are available at the Dryad Digital Repository: https://doi.org/10.5061/dryad.bcc2fqz96 [75].

Authors' contributions. R.T.G. conceived the ideas and designed methodology; R.T.G., D.W.C. and G.H.B. collected the data; I.E.B. and R.T.G. analysed the data; I.E.B. led the writing of the manuscript. All authors gave final approval for publication.

Competing interests. We declare we have no competing interests.

Funding. The project was supported by The Summit Foundation, The University of Florida Foundation, The Mitchell Petersen Foundation, The Ocean Foundation, a key anonymous donor and the Wildlife Conservation Society.

Acknowledgements. We thank our many partner fishers involved in this study, notably Evaristo Muschamp, Darren Castellanos, Alex and Percy Garbutt, Jason Castro and Screech Leslie, and our colleagues Chip Petersen and Shane Young. We are particularly indebted to the Belize Audubon Society who supported fieldwork and lodging; Amigos del Mar, Belize Diving Services, Ramon's Dive Shop, Island Expeditions, Itza Lodge, TMM and Huracan Lodge provided logistical support and provisioning. We further thank the Belize Fisheries Department for extending research permits between 2007 and 2013 when these animals were tagged and tracked.

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
