## [Peer Review File · Royal Society Open Science]

Review History

RSOS-201036.R0 (Original submission)

Review form: Reviewer 1

Is the manuscript scientifically sound in its present form?

No

Are the interpretations and conclusions justified by the results?

No

Is the language acceptable?

No

Do you have any ethical concerns with this paper?

No

Have you any concerns about statistical analyses in this paper?

Yes

Recommendation?

Major revision is needed (please make suggestions in comments)

Comments to the Author(s)

The authors use telemetry data to try and determine space partitioning between age and sex stages of a Caribbean reef shark population and relate this to social factors. The authors clearly have a great dataset that will certainly produce one or more good papers. The problem here is they have attempted to do too much and that has led to weaknesses throughout the whole manuscript.

The biggest problem is the social network analysis. Quantifying social networks from telemetry data is possible but there are many issues that need to be addressed and the data carefully looked at. The authors have used a method that is really not appropriate although there are a few published studies that have (largely incorrectly) done so. The authors simply assume sharks are associating if they are detected at the same receiver during the same minute. The weighted index they use is fine for observation based data (e.g. dolphin groups on the surface) but not appropriate for telemetry. The sharks could well be detected at the receiver and 600 m apart from each other. The authors have just selected an arbitrary time interval to designate a social association. The correct way to produce networks from telemetry data is to search the data for clusters of detections of animals and use the algorithm to classify the grouping (e.g. Gaussian mixture modelling). These methods were developed for birds by Psorakis et al. *Inferring social network structure in ecological systems from spatio-temporal data streams* 2012. *J. Roy. Soc. Interface*.

They were then adapted for use with acoustic telemetry and tested with sharks (Jacoby et al. 2016).

The authors then test their associations with a null model, but the null model presented is not sufficient to test for non-random associations. Farine 2017 (*Methods ecology and evolution*) highlights the issues with incorrect null model selection, and in particular the need for pre-network models. This is particularly an issue when using movement networks to build social networks as there are two stages that need to be examined; the movement stage and the social stage.

Because of these issues, I have no confidence in the social network results presented here and they should be removed or completely redone using more robust methods. I also note the authors routinely describe measuring social interactions; they are measuring associations not interactions which would require observing how the animals are behaving towards each other.

The movement analysis is certainly worthy of publication but despite a lot of analysis, many sections seem somewhat rushed. I would recommend focusing on this section first. For example, despite all the analysis there is no quantitative analysis of habitat separation between sexes or life stages. Rather just qualitative comparisons are given. The network analysis does appear to show life stage differences but no quantitative comparison of space use are made. For example, Stehfest et al. 2015. *Oikos* 124:307-318 combine networks with Markov modelling to come up with population level estimates of what receivers are being used and compare those between the sexes.

The authors also make comparisons about which sex or life stage groups move around more, yet don't provide much quantitative measures. For examples, Heupel and Simpfendorfer have published many papers detailing the methods of how to calculate a roaming index detailing how many receivers animals are detected on. This might make comparisons clearer.

Finally, the authors determine that most movements are random throughout the atoll but this result doesn't make sense for sharks that are using a very small portion of the atoll. If sharks show very restricted space use to a small region of the reef (e.g. the adult females) then movements shouldn't be random, otherwise they would be detected moving across more habitats. Hence the interpretations of the random results do not make sense to me.

Additional comments:

Line 5-8: needs a citation

Line 17: Has there ever been evidence of territoriality in reef sharks?

Line 28: most social networks describe social associations

Line 51-52: How can you do that? I found the introduction very confusing as to the purpose of the study. Was it to look at resource partitioning, social structure and if so why? This is why I feel too much has been done with the data for one paper. Why was the social network analysis performed? Also social associations are not the only (even an important) factor behind resource partitioning. Exploitation competition for example does not require a social interaction. What are the important mechanisms here? For example, are there spatially constrained groups as for some other reef sharks (e.g. Papastamatiou et al. 2018, *Oikos*)? It seems from the results that juvenile sharks use the lagoon more but this isn't even mentioned in the discussion. At the moment the introduction comes across as more a listing of Caribbean reef shark facts.

Line 101-102: The authors cannot assume that sharks that are not detected have died. While overall the species is residential, that doesn't mean individuals don't leave. Similarly, grey reef sharks are a fairly residential species but some are capable of very long distance movements. The second measure of residency (predicted life span of the tags) is more accurate.

Line 135: what associations? Isn't this a movement network?

Line 241-242: What subset? Have these just been arbitrarily picked?

Figure 3: What is the difference in shading? What are the numbers?

Line 328-331: A more likely explanation is that sub adult reef sharks are more likely to disperse. For example Chin et al. 2013 *Marine and Freshwater research*. 23:468-474 show that sub adult blacktip reef sharks leave coastal embayments while juveniles and adults are more residential.

Line 361: If they show restricted space use then how are movements random?

There are also grammatical errors throughout the manuscript.

e.g. line 189-190,

Review form: Reviewer 2

Is the manuscript scientifically sound in its present form?

No

Are the interpretations and conclusions justified by the results?

No

Is the language acceptable?

Yes

Do you have any ethical concerns with this paper?

Yes

Have you any concerns about statistical analyses in this paper?

No

Recommendation?

Major revision is needed (please make suggestions in comments)

Comments to the Author(s)

This study examines the spatial ecology of the Caribbean reef sharks around a remote atoll in Belize. Acoustic tagging data is used to investigate the activity spaces, site fidelity and resident behaviours, across sexes and size classes.

The methods and analytical techniques are fairly routine and the results are generally very interesting and contribute to the knowledge base on ecology of the species sharks. However, there are a number of decisions made within the manuscript that require further justification or investigation (e.g. the exclusion of data or detection thresholds). Further, some of the inferences made and conclusions drawn are speculative or are in contrast to the results of the paper (see comments below).

With some changes, this manuscript will provide a good contribution to the field.

Specific Comments

Abstract line 11 - Ontogenic shifts in activity spaces were found to be insignificant (line 225)

Line 4: "Many" is used 3 times in this sentence, I would suggest rewording.

Line 14 would be better integrated into paragraph starting line 25. Or at least at the end of its existing paragraph to better flow into the next one.

Line 32 - Should be 'discrete'

Line 38 - Such analyses can also be used to guide enforcement strategies within MPAs. See Jacoby et al 2020.

Jacoby, D.M., Ferretti, F., Freeman, R., Carlisle, A.B., Chapple, T.K., Curnick, D.J., Dale, J.J., Schallert, R.J., Tickler, D. and Block, B.A., 2020. Shark movement strategies influence poaching risk and can guide enforcement decisions in a large, remote marine protected area. *Journal of Applied Ecology*, 57(9), pp.1782-1792.

Line 44 - It would like to see the authors give an indication here of how populations at Lighthouse compare with those across the region. Are they monitoring a population at low densities? Or is lighthouse reef more pristine than most places in the region? Furthermore, this should be discussed in the discussion.

Line 62 - I would like to see a reference to support the existence of these spawning sites and possibly indicate their location on Figure 1.

Figure 1 - More detail is needed within the figure legend as it should be standalone from the text. For example, what receivers were deployed and over what time period. Also, a scale bar would be very useful here, along with a key/labelling to better identify reef versus islands.

Line 76 - How variable was this detection range across receivers and between habitat types? I appreciate that they have cited unpublished data here, but reporting some summary statistics here or in the supplementary materials would be very useful.

Line 79 - More detail would be useful here. How many hooks were on each line? For how long were they soaked?

Line 80 - Delete "condition of the shark was noted, and" as this is mentioned in the next sentence.

Line 90 - What is the justification for these selection criteria? And should it be minimum, rather than maximum?

Line 102 – The authors are right to note the issues in calculating residency indices. I would add a reference or two here to justify this approach. Cochran et al 2019 is an useful paper to refer to when discussing these issues. Their approach was to report a Rmin and a Rmax.

Cochran, J.E., Braun, C.D., Cagua, E.F., Campbell Jr, M.F., Hardenstine, R.S., Kattan, A., Priest, M.A., Sinclair-Taylor, T.H., Skomal, G.B., Sultan, S. and Sun, L., 2019. Multi-method assessment of whale shark (*Rhincodon typus*) residency, distribution, and dispersal behavior at an aggregation site in the Red Sea. *PloS one*, 14(9), p.e0222285.

Line 104 – Presumably these were also truncated by the end of the study if the battery life exceeded the study end?

Line 126 – Should be ‘discrete’

Line 129 – Given some of the receivers are much closer together than others, setting a fixed limit of 30 minutes seems oversimplified. What was their justification for using this threshold? The authors could consider parameterising this based on average swim speeds for the species and distances between each receiver.

Line 185 – Given the number of detections by individual is one of the criteria, this should be detailed in table 1. Further, it would be interesting to report data on the 12 not included in the subsequent analyses in the supplementary materials.

Line 188 – It seems there is a typo in here “drove detections drove”.

Line 188 – I would question the exclusion of the data from the two receivers at Blue Hole. Given the purpose of this study is to examine space use across the atoll, it seems odd to exclude a site that is seemingly used regularly by these animals. As the authors currently provide no just reason to exclude these data, I would like to see their inclusion as they could potentially alter the results significantly. If there are sound ecological reasons to exclude them, then these can then be discussed in the paper, comparing the results from the full and subsetted data.

Line 196 – Space needed in ‘arrayare’

Figure 2 – The labels A, B and C are in confusing locations. For instance, the locations of B and C are in the correct sections for males, but not for females. Figure 2 could also be removed, given the lengths are reported in Table 1 and descriptive statistics are outlined in the results section.

Line 208 – Should these be TL?

Line 223 – Given that neither sex nor size class significantly correlated with activity space, I would suggest rewording this sentence to report ranges, rather than suggesting one is larger than another.

Figure 3 – Nice figure. Letters should be capitalised to be consistent with Figure 1. Square boxes need relabelling and a scale bar added (as the figure needs to be standalone).

Line 241 – Provide the number of individuals where home ranges were not calculated.

Table 2 – I would be better to include a column outlining the number of visitations, rather than the asterisks.

Line 288 (and throughout this paragraph) – Not sure the percentages are needed here

Figure 5 – To improve readability for those not familiar with these plots labels (A, B and C) should be added.

Line 318 - *were

Line 328 – This doesn't seem to reflect the results where home ranges/activity spaces were not significant between size classes.

Line 333 - *subadult

Line 339 – It is not clear to me what result this inference is based on.

Line 373 – It would be good to see a bit more discussion about how these findings could be used to support conservation and management of this system. For instance, if fish spawning locations are key for larger females, could these data be used to inform further spatial protection around the atoll, or temporal closures of fishing activity around key spawning events.

Decision letter (RSOS-201036.R0)

Dear Ms Baremore

The Editors assigned to your paper RSOS-201036 "Analysis of resource partitioning by Caribbean reef sharks at a remote atoll in Belize, Central America" have now received comments from reviewers and would like you to revise the paper in accordance with the reviewer comments and any comments from the Editors. Please note this decision does not guarantee eventual acceptance.

Please submit your revised manuscript and required files (see below) no later than 21 days from today's (ie 06-Nov-2020) date. Note: the ScholarOne system will 'lock' if submission of the revision is attempted 21 or more days after the deadline. If you do not think you will be able to meet this deadline please contact the editorial office immediately.

Please note article processing charges apply to papers accepted for publication in Royal Society Open Science (<https://royalsocietypublishing.org/rsos/charges>). Charges will also apply to papers transferred to the journal from other Royal Society Publishing journals, as well as papers submitted as part of our collaboration with the Royal Society of Chemistry

(<https://royalsocietypublishing.org/rsos/chemistry>). Fee waivers are available but must be requested when you submit your revision (<https://royalsocietypublishing.org/rsos/waivers>).

on behalf of Dr Melita Samoilys (Associate Editor) and Pete Smith (Subject Editor)
openscience@royalsociety.org

Reviewer comments to Author:

Reviewer: 1

Comments to the Author(s)

The authors use telemetry data to try and determine space partitioning between age and sex stages of a Caribbean reef shark population and relate this to social factors. The authors clearly have a great dataset that will certainly produce one or more good papers. The problem here is they have attempted to do too much and that has led to weaknesses throughout the whole manuscript.

The biggest problem is the social network analysis. Quantifying social networks from telemetry data is possible but there are many issues that need to be addressed and the data carefully looked at. The authors have used a method that is really not appropriate although there are a few published studies that have (largely incorrectly) done so. The authors simply assume sharks are associating if they are detected at the same receiver during the same minute. The weighted index they use is fine for observation based data (e.g. dolphin groups on the surface) but not appropriate for telemetry. The sharks could well be detected at the receiver and 600 m apart from each other. The authors have just selected an arbitrary time interval to designate a social association. The correct way to produce networks from telemetry data is to search the data for clusters of detections of animals and use the algorithm to classify the grouping (e.g. Gaussian mixture modelling). These methods were developed for birds by Psorakis et al. *Inferring social network structure in ecological systems from spatio-temporal data streams* 2012. *J. Roy. Soc. Interface*.

They were then adapted for use with acoustic telemetry and tested with sharks (Jacoby et al. 2016).

The authors then test their associations with a null model, but the null model presented is not sufficient to test for non-random associations. Farine 2017 (*Methods ecology and evolution*) highlights the issues with incorrect null model selection, and in particular the need for pre-network models. This is particularly an issue when using movement networks to build social networks as there are two stages that need to be examined; the movement stage and the social stage.

Because of these issues, I have no confidence in the social network results presented here and they should be removed or completely redone using more robust methods. I also note the authors routinely describe measuring social interactions; they are measuring associations not interactions which would require observing how the animals are behaving towards each other.

The movement analysis is certainly worthy of publication but despite a lot of analysis, many sections seem somewhat rushed. I would recommend focusing on this section first. For example, despite all the analysis there is no quantitative analysis of habitat separation between sexes or life stages. Rather just qualitative comparisons are given. The network analysis does appear to show

life stage differences but no quantitative comparison of space use are made. For example, Stehfest et al. 2015. *Oikos* 124:307-318 combine networks with Markov modelling to come up with population level estimates of what receivers are being used and compare those between the sexes.

The authors also make comparisons about which sex or life stage groups move around more, yet don't provide much quantitative measures. For examples, Heupel and Simpfendorfer have published many papers detailing the methods of how to calculate a roaming index detailing how many receivers animals are detected on. This might make comparisons clearer.

Finally, the authors determine that most movements are random throughout the atoll but this result doesn't make sense for sharks that are using a very small portion of the atoll. If sharks show very restricted space use to a small region of the reef (e.g. the adult females) then movements shouldn't be random, otherwise they would be detected moving across more habitats. Hence the interpretations of the random results do not make sense to me.

Additional comments:

Line 5-8: needs a citation

Line 17: Has there ever been evidence of territoriality in reef sharks?

Line 28: most social networks describe social associations

Line 51-52: How can you do that? I found the introduction very confusing as to the purpose of the study. Was it to look at resource partitioning, social structure and if so why? This is why I feel too much has been done with the data for one paper. Why was the social network analysis performed? Also social associations are not the only (even an important) factor behind resource partitioning. Exploitation competition for example does not require a social interaction. What are the important mechanisms here? For example, are there spatially constrained groups as for some other reef sharks (e.g. Papastamatiou et al. 2018, *Oikos*)? It seems from the results that juvenile sharks use the lagoon more but this isn't even mentioned in the discussion. At the moment the introduction comes across as more a listing of Caribbean reef shark facts.

Line 101-102: The authors cannot assume that sharks that are not detected have died. While overall the species is residential, that doesn't mean individuals don't leave. Similarly, grey reef sharks are a fairly residential species but some are capable of very long distance movements. The second measure of residency (predicted life span of the tags) is more accurate.

Line 135: what associations? Isn't this a movement network?

Line 241-242: What subset? Have these just been arbitrarily picked?

Figure 3: What is the difference in shading? What are the numbers?

Line 328-331: A more likely explanation is that sub adult reef sharks are more likely to disperse. For example Chin et al. 2013 *Marine and Freshwater research*. 23:468-474 show that sub adult blacktip reef sharks leave coastal embayments while juveniles and adults are more residential.

Line 361: If they show restricted space use then how are movements random?

There are also grammatical errors throughout the manuscript.

e.g. line 189-190,

Reviewer: 2

Comments to the Author(s)

This study examines the spatial ecology of the Caribbean reef sharks around a remote atoll in Belize. Acoustic tagging data is used to investigate the activity spaces, site fidelity and resident behaviours, across sexes and size classes.

The methods and analytical techniques are fairly routine and the results are generally very interesting and contribute to the knowledge base on ecology of the species sharks. However, there are a number of decisions made within the manuscript that require further justification or investigation (e.g. the exclusion of data or detection thresholds). Further, some of the inferences made and conclusions drawn are speculative or are in contrast to the results of the paper (see comments below).

With some changes, this manuscript will provide a good contribution to the field.

Specific Comments

Abstract line 11 – Ontogenic shifts in activity spaces were found to be insignificant (line 225)

Line 4: “Many” is used 3 times in this sentence, I would suggest rewording.

Line 14 would be better integrated into paragraph starting line 25. Or at least at the end of its existing paragraph to better flow into the next one.

Line 32 – Should be ‘discrete’

Line 38 – Such analyses can also be used to guide enforcement strategies within MPAs. See Jacoby et al 2020.

Jacoby, D.M., Ferretti, F., Freeman, R., Carlisle, A.B., Chapple, T.K., Curnick, D.J., Dale, J.J., Schallert, R.J., Tickler, D. and Block, B.A., 2020. Shark movement strategies influence poaching risk and can guide enforcement decisions in a large, remote marine protected area. *Journal of Applied Ecology*, 57(9), pp.1782-1792.

Line 44 – It would like to see the authors give an indication here of how populations at Lighthouse compare with those across the region. Are they monitoring a population at low densities? Or is lighthouse reef more pristine than most places in the region? Furthermore, this should be discussed in the discussion.

Line 62 – I would like to see a reference to support the existence of these spawning sites and possibly indicate their location on Figure 1.

Figure 1 – More detail is needed within the figure legend as it should be standalone from the text. For example, what receivers were deployed and over what time period. Also, a scale bar would be very useful here, along with a key/labelling to better identify reef versus islands.

Line 76 – How variable was this detection range across receivers and between habitat types? I appreciate that they have cited unpublished data here, but reporting some summary statistics here or in the supplementary materials would be very useful.

Line 79 – More detail would be useful here. How many hooks were on each line? For how long were they soaked?

Line 80 – Delete “condition of the shark was noted, and” as this is mentioned in the next sentence.

Line 90 – What is the justification for these selection criteria? And should it be minimum, rather than maximum?

Line 102 – The authors are right to note the issues in calculating residency indices. I would add a reference or two here to justify this approach. Cochran et al 2019 is an useful paper to refer to when discussing these issues. Their approach was to report a R_{min} and a R_{max} .

Cochran, J.E., Braun, C.D., Cagua, E.F., Campbell Jr, M.F., Hardenstine, R.S., Kattan, A., Priest, M.A., Sinclair-Taylor, T.H., Skomal, G.B., Sultan, S. and Sun, L., 2019. Multi-method assessment of whale shark (*Rhincodon typus*) residency, distribution, and dispersal behavior at an aggregation site in the Red Sea. *PloS one*, 14(9), p.e0222285.

Line 104 – Presumably these were also truncated by the end of the study if the battery life exceeded the study end?

Line 126 – Should be ‘discrete’

Line 129 – Given some of the receivers are much closer together than others, setting a fixed limit of 30 minutes seems oversimplified. What was their justification for using this threshold? The authors could consider parameterising this based on average swim speeds for the species and distances between each receiver.

Line 185 – Given the number of detections by individual is one of the criteria, this should be detailed in table 1. Further, it would be interesting to report data on the 12 not included in the subsequent analyses in the supplementary materials.

Line 188 – It seems there is a typo in here “drove detections drove”.

Line 188 – I would question the exclusion of the data from the two receivers at Blue Hole. Given the purpose of this study is to examine space use across the atoll, it seems odd to exclude a site that is seemingly used regularly by these animals. As the authors currently provide no just reason to exclude these data, I would like to see their inclusion as they could potentially alter the results significantly. If there are sound ecological reasons to exclude them, then these can then be discussed in the paper, comparing the results from the full and subsetted data.

Line 196 – Space needed in ‘arrayare’

Figure 2 – The labels A, B and C are in confusing locations. For instance, the locations of B and C are in the correct sections for males, but not for females. Figure 2 could also be removed, given the lengths are reported in Table 1 and descriptive statistics are outlined in the results section.

Line 208 – Should these be TL?

Line 223 – Given that neither sex nor size class significantly correlated with activity space, I would suggest rewording this sentence to report ranges, rather than suggesting one is larger than another.

Figure 3 – Nice figure. Letters should be capitalised to be consistent with Figure 1. Square boxes need relabelling and a scale bar added (as the figure needs to be standalone).

Line 241 – Provide the number of individuals where home ranges were not calculated.

Table 2 – I would be better to include a column outlining the number of visitations, rather than the asterisks.

Line 288 (and throughout this paragraph) – Not sure the percentages are needed here

Figure 5 – To improve readability for those not familiar with these plots labels (A, B and C) should be added.

Line 318 – *were

Line 328 – This doesn’t seem to reflect the results where home ranges/activity spaces were not significant between size classes.

Line 333 - *subadult

Line 339 – It is not clear to me what result this inference is based on.

Line 373 – It would be good to see a bit more discussion about how these findings could be used to support conservation and management of this system. For instance, if fish spawning locations are key for larger females, could these data be used to inform further spatial protection around the atoll, or temporal closures of fishing activity around key spawning events.

===PREPARING YOUR MANUSCRIPT===

===PREPARING YOUR REVISION IN SCHOLARONE===

Author's Response to Decision Letter for (RSOS-201036.R0)

See Appendix A.

RSOS-201036.R1 (Revision)

Review form: Reviewer 1

Is the manuscript scientifically sound in its present form?

Yes

Are the interpretations and conclusions justified by the results?

No

Is the language acceptable?

Yes

Do you have any ethical concerns with this paper?

No

Have you any concerns about statistical analyses in this paper?

No

Recommendation?

Major revision is needed (please make suggestions in comments)

Comments to the Author(s)

I thank the authors for their efforts in addressing both reviewers. Now that the social network analysis has been removed, and more movement analysis has been performed, the manuscript looks much better. I do still see some issues however that should be addressed.

Introduction:

Pg 3, Line 19: some basic description of ecotourism should be defined here especially for readers who are not familiar with marine ecotourism. Are these baited dives? How are movements disrupted?

Pg 3, Line 34-35: There are several studies that have tracked reef sharks over multiple years in pristine systems including Seychelles (Lea et al. 2016, Proc B R Soc), Palmyra (Papastamatiou et al. 2010 J exp mar boil and ecol, 2018, Oikos), great barrier reef (Heupel et al's many papers).

Pg 4, line 23-24: This gets confusing as some of these studies were movement networks and some were social networks. As social networks are not used in the study I would stick to movement networks. Also note the social network studies did not look at interactions, only associations so the sentence is incorrect.

Pg 4, line 28: I don't understand what the authors mean here. What is unique about shark movements that cant be analyzed using traditional methods? The same methods used for sharks are used for all marine animals?

Pg 5, line 48: Much more detailed movement data of Caribbean reef sharks in the Bahamas now exists. See Gallagher et al. 2021, Frontiers in Marine Science

Pg 6, line 37-42. Remove the actual CPUE values. This looks odd in the introduction and doesn't add any needed details.

Pg 6, line 55: You don't determine how intra-specific interactions drive space use. You never measure interactions. Do you mean ontogenetic patterns in space use and movements?

Methods:

Pg 9: Where were sharks tagged? Its very important to note if sharks were tagged in one place (e.g. south portion of the atoll). If adult females have restricted space use, then there may be other individuals using other portions of the atoll that have just not been tagged. This is particularly important for adult females which are very residential. Has the analysis been biased by sampling location? This wouldn't detract from the paper but needs to be addressed.

Pg 10, line 26-28: It makes no sense to claim that sharks that are not detected have either died or had transmitter failure. The other obvious reason is they left the atoll. No reef shark species is 100% residential. The authors even prove this point in the discussion by noting that one shark was detected 90km away! How can they not consider dispersal from the atoll a possibility?

Pg 11, line 12-14: Isn't another explanation for the short residency times that sharks don't stay all that long at the atoll? I don't understand why the authors believe residency is so high, when their own data suggests it isn't?

Pg 24, line 32-33: There is a typo in the residency. Is that 0.001?

Results: I don't fully understand how adult female UD estimates suggest that sharks are only using the south portion of the atoll, but the Markov chain analysis suggests they are transiting along the western side of the atoll? If they are detected transiting shouldn't there be some space use at other parts of the atoll?

Discussion

Pg 32, line 27-33. The authors note that there was less residency and more roaming for intermediate sized females as opposed to juveniles or adults. This matches with other studies showing dispersal of the intermediate size ranged animals. However, then the authors state there is a linear trend between shark size and residency and home range size. If it's the intermediate sized animals moving the most, how can the trend be linear?

Line 39-41: The authors own evidence of dispersal of sharks to other atolls

Pg 33, line 39-42. Papastamatiou et al. 2020 Proc R Soc B is a better reference for social groupings in grey reef sharks. It also showed less evidence of associations by sex.

General comments: the authors use residency and site fidelity interchangeably but they are different things. Residency implies an animal spends most of its time at a location. Site fidelity is an animal that returns to a specific area, but spends more time away from that area. Please make sure the right term is used throughout.

Decision letter (RSOS-201036.R1)

Dear Ms Baremore

The Editors assigned to your paper RSOS-201036.R1 "Movements and residency of Caribbean reef sharks at a remote atoll in Belize, Central America" have now received comments from reviewers and would like you to revise the paper in accordance with the reviewer comments and any comments from the Editors. Please note this decision does not guarantee eventual acceptance.

Please submit your revised manuscript and required files (see below) no later than 21 days from today's (ie 14-May-2021) date. Note: the ScholarOne system will 'lock' if submission of the

revision is attempted 21 or more days after the deadline. If you do not think you will be able to meet this deadline please contact the editorial office immediately.

on behalf of Dr Melita Samoilyis (Associate Editor) and Pete Smith (Subject Editor)
openscience@royalsociety.org

Reviewer comments to Author:

Reviewer: 1

Comments to the Author(s)

I thank the authors for their efforts in addressing both reviewers. Now that the social network analysis has been removed, and more movement analysis has been performed, the manuscript looks much better. I do still see some issues however that should be addressed.

Introduction:

Pg 3, Line 19: some basic description of ecotourism should be defined here especially for readers who are not familiar with marine ecotourism. Are these baited dives? How are movements disrupted?

Pg 3, Line 34-35: There are several studies that have tracked reef sharks over multiple years in pristine systems including Seychelles (Lea et al. 2016, Proc B R Soc), Palmyra (Papastamatiou et al. 2010 J exp mar boil and ecol, 2018, Oikos), great barrier reef (Heupel et al's many papers).

Pg 4, line 23-24: This gets confusing as some of these studies were movement networks and some were social networks. As social networks are not used in the study I would stick to movement networks. Also note the social network studies did not look at interactions, only associations so the sentence is incorrect.

Pg 4, line 28: I don't understand what the authors mean here. What is unique about shark movements that can't be analyzed using traditional methods? The same methods used for sharks are used for all marine animals?

Pg 5, line 48: Much more detailed movement data of Caribbean reef sharks in the Bahamas now exists. See Gallagher et al. 2021, Frontiers in Marine Science

Pg 6, line 37-42. Remove the actual CPUE values. This looks odd in the introduction and doesn't add any needed details.

Pg 6, line 55: You don't determine how intra-specific interactions drive space use. You never measure interactions. Do you mean ontogenetic patterns in space use and movements?

Methods:

Pg 9: Where were sharks tagged? Its very important to note if sharks were tagged in one place (e.g. south portion of the atoll). If adult females have restricted space use, then there may be other individuals using other portions of the atoll that have just not been tagged. This is particularly

important for adult females which are very residential. Has the analysis been biased by sampling location? This wouldn't detract from the paper but needs to be addressed.

Pg 10, line 26-28: It makes no sense to claim that sharks that are not detected have either died or had transmitter failure. The other obvious reason is they left the atoll. No reef shark species is 100% residential. The authors even prove this point in the discussion by noting that one shark was detected 90km away! How can they not consider dispersal from the atoll a possibility?

Pg 11, line 12-14: Isn't another explanation for the short residency times that sharks don't stay all that long at the atoll? I don't understand why the authors believe residency is so high, when their own data suggests it isn't?

Pg 24, line 32-33: There is a typo in the residency. Is that 0.001?

Results: I don't fully understand how adult female UD estimates suggest that sharks are only using the south portion of the atoll, but the Markov chain analysis suggests they are transiting along the western side of the atoll? If they are detected transiting shouldn't there be some space use at other parts of the atoll?

Discussion

Pg 32, line 27-33. The authors note that there was less residency and more roaming for intermediate sized females as opposed to juveniles or adults. This matches with other studies showing dispersal of the intermediate size ranged animals. However, then the authors state there is a linear trend between shark size and residency and home range size. If it's the intermediate sized animals moving the most, how can the trend be linear?

Line 39-41: The authors own evidence of dispersal of sharks to other atolls

Pg 33, line 39-42. Papastamatiou et al. 2020 Proc R Soc B is a better reference for social groupings in grey reef sharks. It also showed less evidence of associations by sex.

General comments: the authors use residency and site fidelity interchangeably but they are different things. Residency implies an animal spends most of its time at a location. Site fidelity is an animal that returns to a specific area, but spends more time away from that area. Please make sure the right term is used throughout.

===PREPARING YOUR MANUSCRIPT===

If you have been asked to revise the written English in your submission as a condition of publication, you must do so, and you are expected to provide evidence that you have received

language editing support. The journal would prefer that you use a professional language editing service and provide a certificate of editing, but a signed letter from a colleague who is a native speaker of English is acceptable. Note the journal has arranged a number of discounts for authors using professional language editing services (<https://royalsociety.org/journals/authors/benefits/language-editing/>).

===PREPARING YOUR REVISION IN SCHOLARONE===

<https://royalsociety.org/journals/authors/author-guidelines/#supplementary-material> to

include a suitable title and informative caption. An example of appropriate titling and captioning may be found at https://figshare.com/articles/Table_S2_from_Is_there_a_trade-off_between_peak_performance_and_performance_breadth_across_temperatures_for_aerobic_sc_ope_in_teleost_fishes_/3843624.

Author's Response to Decision Letter for (RSOS-201036.R1)

See Appendix B.

Decision letter (RSOS-201036.R2)

Dear Ms Baremore,

On behalf of the Editors, we are pleased to inform you that your Manuscript RSOS-201036.R2 "Movements and residency of Caribbean reef sharks at a remote atoll in Belize, Central America" has been accepted for publication in Royal Society Open Science subject to minor revision in accordance with the referees' reports. Please find the referees' comments along with any feedback from the Editors below my signature.

Please submit your revised manuscript and required files (see below) no later than 7 days from today's (ie 05-Jul-2021) date. Note: the ScholarOne system will 'lock' if submission of the revision is attempted 7 or more days after the deadline. If you do not think you will be able to meet this deadline please contact the editorial office immediately.

on behalf of Dr Melita Samoilys (Associate Editor) and Pete Smith (Subject Editor)
 openscience@royalsociety.org

Associate Editor Comments to Author (Dr Melita Samoilys):

The paper presents an important tagging study over ~ 7 years of 77 endangered Caribbean reef sharks, providing valuable information on space use and movements at an atoll in Belize. Figures 5 and 7 capture the results very well. The analyses are robust and the results contribute significantly to our understanding of this important and threatened species as well as reef sharks generally. The paper is almost ready for publication save for some minor edits and suggestions for moving figures and tables to Supplementary material to give the main text better focus and to save space.

Small final edits:

1. Table 1 should be ordered by a more interesting column than ID number. This number is not that interesting to the reader. So, for example total number of days detected would be interesting, and perhaps M and F separated into two parts of the table.
2. page 26. FSA should be spelt out in full with some explanation of what it is - early in the paper perhaps when referring to Figure 1, and "reef fish" added when referring to "spawning sites" to clarify in the Discussion on page 26-27.
3. Page 27, Line 378. This text "...many more were captured but unreported (R. Graham, pers. obs.)" is not clear. Many more than what? Perhaps this should be referring to the 10 tagged sharks that were never detected again? or of the 77 that were barely detected, say for less than 30 days?
4. page 28, lines 397-399> this sentence is not clear using terms that have not been presented previously "... and social networks of sharks have shown to be robust to the removal of nodes with high degree and low betweenness". Please explain better or remove.
5. In the interests of space regarding tables and figures, and because there are too many of them: Table 1 could go in suppl. material, or if retained in the main text then Fig 2 should go to Suppl. material.
 Fig 4. be considered for Suppl. material.
 Fig. 6 to suppl. material while Table 4 in the main text.
 Table 5 to suppl. since the significant receivers are shown in Figure 7.

===PREPARING YOUR MANUSCRIPT===

Your revised paper should include the changes requested by the referees and Editors of your manuscript. You should provide two versions of this manuscript and both versions must be provided in an editable format:
 one version identifying all the changes that have been made (for instance, in coloured highlight, in bold text, or tracked changes);
 a 'clean' version of the new manuscript that incorporates the changes made, but does not highlight them. This version will be used for typesetting.

===PREPARING YOUR REVISION IN SCHOLARONE===

- If you are providing image files for potential cover images, please upload these at this step, and inform the editorial office you have done so. You must hold the copyright to any image provided.
- A copy of your point-by-point response to referees and Editors. This will expedite the preparation of your proof.

- Ensure that your data access statement meets the requirements at <https://royalsociety.org/journals/authors/author-guidelines/#data>. You should ensure that you cite the dataset in your reference list. If you have deposited data etc in the Dryad repository, please only include the 'For publication' link at this stage. You should remove the 'For review' link.
- If you are requesting an article processing charge waiver, you must select the relevant waiver option (if requesting a discretionary waiver, the form should have been uploaded at Step 3 'File upload' above).
- If you have uploaded ESM files, please ensure you follow the guidance at <https://royalsociety.org/journals/authors/author-guidelines/#supplementary-material> to include a suitable title and informative caption. An example of appropriate titling and captioning may be found at https://figshare.com/articles/Table_S2_from_Is_there_a_trade-off_between_peak_performance_and_performance_breadth_across_temperatures_for_aerobic_scorpions_in_teleost_fishes_/3843624.

Author's Response to Decision Letter for (RSOS-201036.R2)

See Appendix C.

Decision letter (RSOS-201036.R3)

Dear Ms Baremore,

I am pleased to inform you that your manuscript entitled "Movements and residency of Caribbean reef sharks at a remote atoll in Belize, Central America" is now accepted for publication in Royal Society Open Science.

on behalf of Dr Melita Samoilys (Associate Editor) and Pete Smith (Subject Editor)
openscience@royalsociety.org

Appendix A

Dear Editor and reviewers,

We are grateful for the thorough and careful recommendations provided by the reviewers. We have reanalyzed our data as suggested and revised the manuscript as appropriate. Please find responses to reviewers' comments and suggestions below each comment in blue. In cases where comments were no longer relevant due to changes in the manuscript, we reply with 'NA.'

Please find detailed responses below:

Comments to the Author(s)

The authors use telemetry data to try and determine space partitioning between age and sex stages of a Caribbean reef shark population and relate this to social factors. The authors clearly have a great dataset that will certainly produce one or more good papers. The problem here is they have attempted to do too much and that has led to weaknesses throughout the whole manuscript.

The biggest problem is the social network analysis. Quantifying social networks from telemetry data is possible but there are many issues that need to be addressed and the data carefully looked at. The authors have used a method that is really not appropriate although there are a few published studies that have (largely incorrectly) done so. The authors simply assume sharks are associating if they are detected at the same receiver during the same minute. The weighted index they use is fine for observation based data (e.g. dolphin groups on the surface) but not appropriate for telemetry. The sharks could well be detected at the receiver and 600 m apart from each other. The authors have just selected an arbitrary time interval to designate a social association. The correct way to produce networks from telemetry data is to search the data for clusters of detections of animals and use the algorithm to classify the grouping (e.g. Gaussian mixture modelling). These methods were developed for birds by Psorakis et al. Inferring social network structure in ecological systems from spatio-temporal data streams 2012. J. Roy. Soc. Interface. They were then adapted for use with acoustic telemetry and tested with sharks (Jacoby et al. 2016).

The authors then test their associations with a null model, but the null model presented is not sufficient to test for non-random associations. Farine 2017 (Methods ecology and evolution) highlights the issues with incorrect null model selection, and in particular the need for pre-network models. This is particularly an issue when using movement networks to build social networks as there are two stages that need to be examined; the movement stage and the social stage.

Because of these issues, I have no confidence in the social network results presented here and they should be removed or completely redone using more robust methods. I also note the authors routinely describe measuring social interactions; they are measuring associations not interactions which would require observing how the animals are behaving towards each other.

Thank you for your thoughtful analysis and recommendations. We agree that the scope of the paper is maybe a bit too broad, and therefore will benefit from a more focused group of analyses. We removed the social analysis sections of the manuscript and will revisit the analyses for future work.

The movement analysis is certainly worthy of publication but despite a lot of analysis, many sections seem somewhat rushed. I would recommend focusing on this section first. For example, despite all the analysis there is no quantitative analysis of habitat separation between sexes or life stages. Rather just qualitative comparisons are given. The network analysis does appear to show life stage differences but no quantitative comparison of space use are made. For example, Stehfest et al. 2015. Oikos 124:307-318 combine networks with Markov modelling to come up with population level estimates of what receivers are being used and compare those between the sexes.

The authors also make comparisons about which sex or life stage groups move around more, yet don't provide much quantitative measures. For examples, Heupel and Simpfendorfer have published many papers detailing the methods of how to calculate a roaming index detailing how many receivers animals are detected on. This might make comparisons clearer.

We re-analyzed the data using Stehfest's 'sharkov' model and code by sex and life history stage. Probability matrices were built and the eigenvector scores used to investigate differences.

We calculated a Roaming Index calculation and associated figures. We have also added a few linear model analyses to test for differences in sex and size.

Finally, the authors determine that most movements are random throughout the atoll but this result doesn't make sense for sharks that are using a very small portion of the atoll. If sharks show very restricted space use to a small region of the reef (e.g. the adult females) then movements shouldn't be random, otherwise they would be detected moving across more habitats. Hence the interpretations of the random results do not make sense to me.

We have removed this analysis from the manuscript, instead focusing on the Markov chain model

Additional comments:
Line 5-8: needs a citation

Added

Line 17: Has there ever been evidence of territoriality in reef sharks?

This is a good point, and I suppose is something that many researchers postulate without actually testing. The original statement of the previous authors' hypothesis is left in, but with the caveat from Papastamatiou et al's 2018 paper on spatial segregation.

Line 28: most social networks describe social associations

Line 51-52: How can you do that? I found the introduction very confusing as to the purpose of the study. Was it to look at resource partitioning, social structure and if so why? This is why I feel too much has been done with the data for one paper. Why was the social network analysis performed? Also social associations are not the only (even an important) factor behind resource partitioning. Exploitation competition for example does not require a social interaction. What are the important mechanisms here? For example, are there spatially constrained groups as for some other reef sharks (e.g. Papastamatiou et al. 2018, Oikos)? It seems from the results that juvenile sharks use the lagoon more but this isn't even mentioned in the discussion. At the moment the introduction comes across as more a listing of Caribbean reef shark facts.

The introduction has been re-worked to better fit the objectives of the manuscript.

Line 101-102: The authors cannot assume that sharks that are not detected have died. While overall the species is residential, that doesn't mean individuals don't leave. Similarly, grey reef sharks are a fairly residential species but some are capable of very long distance movements. The second measure of residency (predicted life span of the tags) is more accurate.

Following this and recommendations from reviewer 2, we have added a justification for reporting both RI based on tag life and Rmax based on last detection. Statistical analyses were performed using RI, though one plot includes Rmax.

Line 135: what associations? Isn't this a movement network?

This has been removed

Line 241-242: What subset? Have these just been arbitrarily picked?

This has been removed

Figure 3: What is the difference in shading? What are the numbers?

The figure caption has been amended

Line 328-331: A more likely explanation is that sub adult reef sharks are more likely to disperse. For example Chin et al. 2013 Marine and Freshwater research. 23:468-474 show that sub adult blacktip reef sharks leave coastal embayments while juveniles and adults are more residential.

The language was amended to suggest dispersal rather than increased activity space

Line 361: If they show restricted space use then how are movements random?

NA in the revision

There are also grammatical errors throughout the manuscript.e.g. line 189-190,

The manuscript was carefully re-read after revisions to minimize grammatical errors.

Reviewer: 2

Comments to the Author(s)

This study examines the spatial ecology of the Caribbean reef sharks around a remote atoll in Belize. Acoustic tagging data is used to investigate the activity spaces, site fidelity and resident behaviours, across sexes and size classes.

The methods and analytical techniques are fairly routine and the results are generally very interesting and contribute to the knowledge base on ecology of the species sharks. However, there are a number of decisions made within the manuscript that require further justification or investigation (e.g. the exclusion of data or detection thresholds). Further, some of the inferences made and conclusions drawn are speculative or are in contrast to the results of the paper (see comments below).

With some changes, this manuscript will provide a good contribution to the field.

Specific Comments

Abstract line 11 – Ontogenic shifts in activity spaces were found to be insignificant (line 225)

NA in the revision

Line 4: “Many” is used 3 times in this sentence, I would suggest rewording.

Many thanks for catching this, the sentence has been reworded.

Line 14 would be better integrated into paragraph starting line 25. Or at least at the end of its existing paragraph to better flow into the next one.

The paragraphs have been re-organized for clarity and flow.

Line 32 – Should be ‘discrete’

Reworded.

Line 38 – Such analyses can also be used to guide enforcement strategies within MPAs. See Jacoby et al 2020.

Jacoby, D.M., Ferretti, F., Freeman, R., Carlisle, A.B., Chapple, T.K., Curnick, D.J., Dale, J.J., Schallert, R.J., Tickler, D. and Block, B.A., 2020. Shark movement strategies influence poaching risk and can guide enforcement decisions in a large, remote marine protected area. *Journal of Applied Ecology*, 57(9), pp.1782-1792.

This has been added to the text, and the paper is a valuable contribution.

Line 44 – It would like to see the authors give an indication here of how populations at Lighthouse compare with those across the region. Are they monitoring a population at low densities? Or is lighthouse reef more pristine than most places in the region? Furthermore, this should be discussed in the discussion.

Text has been amended in the introduction and discussion.

Line 62 – I would like to see a reference to support the existence of these spawning sites and possibly indicate their location on Figure 1.

Figure 1 – More detail is needed within the figure legend as it should be standalone from the text. For example, what receivers were deployed and over what time period. Also, a scale bar would be very useful here, along with a key/labelling to better identify reef versus islands.

Figure 1 has been updated to include the requested information.

Line 76 – How variable was this detection range across receivers and between habitat types? I appreciate that they have cited unpublished data here, but reporting some summary statistics here or in the supplementary materials would be very useful.

Line 79 – More detail would be useful here. How many hooks were on each line? For how long were they soaked?

More detail has been added.

Line 80 – Delete “condition of the shark was noted, and” as this is mentioned in the next sentence.

Deleted

Line 90 – What is the justification for these selection criteria? And should it be minimum, rather than maximum?

Language has been added to justify this and the sentence is slightly reworded.

Line 102 – The authors are right to note the issues in calculating residency indices. I would add a reference or two here to justify this approach. Cochran et al 2019 is an useful paper to refer to when discussing these issues. Their approach was to report a Rmin and a Rmax.

Cochran, J.E., Braun, C.D., Cagua, E.F., Campbell Jr, M.F., Hardenstine, R.S., Kattan, A., Priest, M.A., Sinclair-Taylor, T.H., Skomal, G.B., Sultan, S. and Sun, L., 2019. Multi-method assessment of whale shark (*Rhincodon typus*) residency, distribution, and dispersal behavior at an aggregation site in the Red Sea. *PLoS one*, 14(9), p.e0222285.

Thank you for this reference, we have updated the table and text accordingly

Line 104 – Presumably these were also truncated by the end of the study if the battery life exceeded the study end?

Correct, we have clarified this

Line 126 – Should be ‘discrete’

Changed

Line 129 – Given some of the receivers are much closer together than others, setting a fixed limit of 30 minutes seems oversimplified. What was their justification for using this threshold? The authors could consider parameterising this based on average swim speeds for the species and distances between each receiver.

We have changed the time step to one hour, and added language about the average swimming speed of similar species.

Line 185 – Given the number of detections by individual is one of the criteria, this should be detailed in table 1. Further, it would be interesting to report data on the 12 not included in the subsequent analyses in the

supplementary materials.

The criteria were slightly modified – as detailed in the methods, and a supplementary table of all tagged sharks was added.

Line 188 – It seems there is a typo in here “drove detections drove”.

NA in the revision

Line 188 – I would question the exclusion of the data from the two receivers at Blue Hole. Given the purpose of this study is to examine space use across the atoll, it seems odd to exclude a site that is seemingly used regularly by these animals. As the authors currently provide no just reason to exclude these data, I would like to see their inclusion as they could potentially alter the results significantly. If there are sound ecological reasons to exclude them, then these can then be discussed in the paper, comparing the results from the full and subsetted data.

We agree with this point, and have re-run all analyses with the blue hole receivers. The text and figures have been updated accordingly, and the implications are discussed.

Line 196 – Space needed in ‘arrayare’

Thank you, corrected.

Figure 2 – The labels A, B and C are in confusing locations. For instance, the locations of B and C are in the correct sections for males, but not for females. Figure 2 could also be removed, given the lengths are reported in Table 1 and descriptive statistics are outlined in the results section.

Figure 2 has been removed.

Line 208 – Should these be TL?

Yes they should, though the section has been rewritten.

Line 223 – Given that neither sex nor size class significantly correlated with activity space, I would suggest rewording this sentence to report ranges, rather than suggesting one is larger than another.

Figure 3 – Nice figure. Letters should be capitalised to be consistent with Figure 1. Square boxes need relabelling and a scale bar added (as the figure needs to be standalone).

Thank you, the figure has been reworked with the Blue Hole receivers added and we included your suggestions in the revision.

Line 241 – Provide the number of individuals where home ranges were not calculated.

No longer relevant.

Table 2 – I would be better to include a column outlining the number of visitations, rather than the asterisks.

NA

Line 288 (and throughout this paragraph) – Not sure the percentages are needed here

NA

Figure 5 – To improve readability for those not familiar with these plots labels (A, B and C) should be added.

NA

Line 318 - *were

NA

Line 328 – This doesn't seem to reflect the results where home ranges/activity spaces were not significant between size classes.

Amended

Line 333 - *subadult

Line 339 – It is not clear to me what result this inference is based on.

This has been rewritten to reflect observed occurrences of dispersion by juvenile reef sharks

Line 373 – It would be good to see a bit more discussion about how these findings could be used to support conservation and management of this system. For instance, if fish spawning locations are key for larger females, could these data be used to inform further spatial protection around the atoll, or temporal closures of fishing activity around key spawning events.

This has been added to the discussion.

Appendix B

I thank the authors for their efforts in addressing both reviewers. Now that the social network analysis has been removed, and more movement analysis has been performed, the manuscript looks much better. I do still see some issues however that should be addressed.

Introduction:

Pg 3, Line 19: some basic description of ecotourism should be defined here especially for readers who are not familiar with marine ecotourism. Are these baited dives? How are movements disrupted?

Language has been added to this paragraph to describe ecotourism as related to diving and sharks in Belize (page 3, line 7).

Pg 3, Line 34-35: There are several studies that have tracked reef sharks over multiple years in pristine systems including Seychelles (Lea et al. 2016, Proc B R Soc), Palmyra (Papastamatiou et al. 2010 J exp mar boil and ecol, 2018, Oikos), great barrier reef (Heupel et al's many papers).

These references have been added, though we do note that few of these reported robust numbers across all stages of the species' life history and therefore the sentence has remained unchanged.

Pg 4, line 23-24: This gets confusing as some of these studies were movement networks and some were social networks. As social networks are not used in the study I would stick to movement networks. Also note the social network studies did not look at interactions, only associations so the sentence is incorrect.

This has been corrected.

Pg 4, line 28: I don't understand what the authors mean here. What is unique about shark movements that cant be analyzed using traditional methods? The same methods used for sharks are used for all marine animals?

This sentence has been changed from 'traditional' to 'observational.'

Pg 5, line 48: Much more detailed movement data of Caribbean reef sharks in the Bahamas now exists. See Gallagher et al. 2021, Frontiers in Marine Science

This reference and its relevant information have been added.

Pg 6, line 37-42. Remove the actual CPUE values. This looks odd in the introduction and doesn't add any needed details.

These have been removed.

Pg 6, line 55: You don't determine how intra-specific interactions drive space use. You never measure interactions. Do you mean ontogenetic patterns in space use and movements?

The reviewer is correct, and this has been reworded.

Methods:

Pg 9: Where were sharks tagged? Its very important to note if sharks were tagged in one place (e.g. south portion of the atoll). If adult females have restricted space use, then there may be other individuals using other portions of the atoll that have just not been tagged. This is particularly important for adult females which are very residential. Has the analysis been biased by sampling location? This wouldn't detract from the paper but needs to be addressed.

Tagging locations have been added to the map on Figure 1 and a brief explanation of tagging sites was added to the text.

Pg 10, line 26-28: It makes no sense to claim that sharks that are not detected have either died or had transmitter failure. The other obvious reason is they left the atoll. No reef shark species is 100% residential. The authors even prove this point in the discussion by nothing that one shark was detected 90km away! How can they not consider dispersal from the atoll a possibility? Pg 11, line 12-14: Isnt another explanation for the short residency times that sharks don't stay all that long at the atoll? I don't understand why the authors believe residency is so high, when their own data suggests it isn't?

This sentence was removed.

Pg 24, line 32-33: There is a typo in the residency. Is that 0.001?

The residency values have been checked.

Results: I dont fully understand how adult female UD estimates suggest that sharks are only using the south portion of the atoll, but the Markov chain analysis suggests they are transiting along the western side of the atoll? If they are detected transiting shouldn't there be some space use at other parts of the atoll?

We thank the reviewer for pointing this out, and have re-examined the data to ensure that our analysis was correct. Upon further examination, we found that the detections at receivers 18 and 19 were somewhat anomalous for one large female. We applied a more stringent criteria for removing false detections for all sharks, which resulted in a more reasonable transition matrix that supports the UD analyses. Language has been added to reflect this and the relevant figures and tables have been updated. All analyses were re-run after the data were re-screened for false detections, but there were no significant changes to the results, aside from the transition corridors.

Discussion

Pg 32, line 27-33. The authors note that there was less residency and more roaming for intermediate sized females as opposed to juveniles or adults. This matches with other studies showing dispersal of the intermediate size ranged animals. However, then the authors state there is a linear trend between shark size and residency and home range size. If it's the intermediate sized animals moving the most, how can the trend be linear?

This was reworded to reflect the analyses done.

Line 39-41: The authors own evidence of dispersal of sharks to other atolls

Pg 33, line 39-42. Papastamatiou et al. 2020 Proc R Soc B is a better reference for social groupings in grey reef sharks. It also showed less evidence of associations by sex.

This reference and its relevant information have been added.

General comments: the authors use residency and site fidelity interchangeably but they are different things. Residency implies an animal spends most of its time at a location. Site fidelity is an animal that returns to a specific area, but spends more time away from that area. Please make sure the right term is used throughout.

We thank the reviewer for pointing this out, and we have made corrections in the text.

Appendix C

We thank the Editor for their thoughtful comments on this manuscript. We have made all of the suggested changes. Please see our responses to each suggestion in blue below:

Small final edits:

1. Table 1 should be ordered by a more interesting column than ID number. This number is not that interesting to the reader. So, for example total number of days detected would be interesting, and perhaps M and F separated into two parts of the table.

We have sorted the table by sex, and then by total number of days detected.

2. page 26. FSA should be spelt out in full with some explanation of what it is - early in the paper perhaps when referring to Figure 1, and “reef fish” added when referring to “spawning sites” to clarify in the Discussion on page 26-27.

We have defined FSA the methods section (line 84) and added reef fish to the Discussion (line 280).

3. Page 27, Line 378. This text “...many more were captured but unreported (R. Graham, pers. obs.)” is not clear. Many more than what? Perhaps this should be referring to the 10 tagged sharks that were never detected again? or of the 77 that were barely detected, say for less than 30 days?

We have reworded the text to say “...more were captured as verified by photos of external tags but were unreported by fishers and therefore could not be identified (R. Graham, pers. obs.)”, and hope that this is more easily understood.

4. page 28, lines 397-399> this sentence is not clear using terms that have not been presented previously “... and social networks of sharks have shown to be robust to the removal of nodes with high degree and low betweenness”. Please explain better or remove.

We have reworded this to say “and social networks of sharks have shown to be robust to the removal of highly connected individuals,” which we hope will be more clear to the reader.

5. In the interests of space regarding tables and figures, and because there are too many of them: Table 1 could go in suppl. material, or if retained in the main text then Fig 2 should go to Suppl. material.

We kept Table 1 in the main text as we feel that it contains information that the reader may want to access quickly, and have made Fig 2 a supplementary figure.

Fig 4. be considered for Suppl. material.

Figure 4 was included as a supplemental figure

Fig. 6 to suppl. material while Table 4 in the main text.

Figure 6 was included as a supplementary figure and Table 4 was retained in the main text.

Table 5 to suppl. since the significant receivers are shown in Figure 7.

Table 5 was included as a supplementary table.